# Imagine a City: CityGenAgent for Procedural 3D City Generation

## Abstract

The automated generation of interactive 3D cities is a critical challenge with broad applications in autonomous driving, virtual reality, and embodied intelligence. While recent advances in generative models and procedural techniques have improved the realism and scalability of city generation, existing methods often struggle with high-fidelity asset creation, controllability, and manipulation. In this work, we introduce CityGenAgent, a natural language-driven framework for hierarchical procedural generation of high-quality 3D cities. Our approach introduces two core programs—**Block Program** and **Building Program**—which decompose city generation into interpretable and editable components. To ensure structural correctness and semantic alignment, we adopt a two-stage learning strategy: (1) Supervised Fine-Tuning (SFT). We train BlockGen and BuildingGen to generate valid programs that adhere to schema constraints, including non-self-intersecting polygons and complete fields; (2) Reinforcement Learning (RL). We introduce Spatial Alignment Reward for accurate spatial reasoning and Visual Consistency Reward to bridge the gap between textual descriptions and 3D realizations. Benefiting from program-based representation and the models' generalization, CityGenAgent supports natural language editing and manipulation. Comprehensive evaluations demonstrate superior semantic alignment, visual quality, and controllability compared to existing methods, establishing a robust foundation for scalable 3D city generation. Demos are available at our project page.

## 1 Introduction

The development of interactive world models has emerged as a key research area (Team et al., 2025), which fundamentally depending on advances in 3D scene generation techniques. Among these, urban environments are particularly important, with wide-ranging applications in autonomous driving, game asset development, virtual reality and embodied intelligence. However, cities are composed of complex road networks, diverse building structures, and various urban facilities, all of which pose challenges for automated generation and realistic simulation.

Procedural generation, which employs rule-based and algorithmic methods to create high-quality 3D content, has a long history in video games and computer graphics. Traditional approaches (Parish & Müller, 2001; Kelly & McCabe, 2007) employ rule-based systems to generate road networks and buildings but they require considerable manual intervention and significant labor expenses. The notable progress in deep learning in recent years has led to significant advances in rendering-based methods (Lin et al., 2023; Xie et al., 2024; Shen et al., 2022), which can synthesize realistic images of urban scenes, but they still lack consistent and precise 3D geometry, which limits practical usage.

Leveraging the prior knowledge and reasoning capabilities of Large Language Models (LLMs), recent studies have attempted to integrate LLMs with procedural generation techniques to enhance the output quality and structural richness of generated environments, for example, in natural scenes (Duan et al., 2025; Sun et al., 2025a) and indoor scenes (Feng et al., 2023; Yang et al., 2024). Recent studies have demonstrated that integrating LLMs with procedural techniques can bring notable progress in city generation, for example, CityCraft (Deng et al., 2024), UrbanWorld (Shang et al., 2024), and CityX (Zhang et al., 2024), showing the potential of LLMs in this domain. Nevertheless, these methods rely mainly on general-purpose LLMs that lack domain-specific knowledge

of urban design, and they remain constrained by their reliance on fixed assets, which limits the creativity and flexibility of generation. Moreover, they often struggle to deliver high-quality results in terms of layout coherence, geometric fidelity, and texture realism, leaving a considerable gap from the requirements of realistic and scalable 3D cities.

In this paper, we propose **CityGenAgent**, a natural language-driven framework for hierarchical procedural generation of high-quality 3D cities. At the core of our approach are two domain-specific language (DSL) programs, the **Block Program** and the **Building Program**, which provide a two-level decomposition and parameterization of cities. These programs offer a compact yet expressive representation: a city block layout and a building structure encoded by a simple set of parameters, which not only facilitates efficient generation but also enables controllable editing and manipulation.

Built upon these programs, we present two specialized agents—**BlockGen** and **BuildingGen**—which are trained via Supervised Fine-Tuning (SFT) and Reinforcement Learning (RL) to ensure the generated 3D structures are both spatially plausible and visually faithful to their programmatic specifications. BlockGen focuses on generating coherent block layouts, learning to place buildings and urban elements in a physically plausible manner that avoids collisions and maintains reasonable density. To this end, we design a **Spatial Alignment Reward** that evaluates semantic consistency with textual descriptions and enforces structural plausibility, enabling BlockGen to generalize beyond limited synthetic data. BuildingGen, in turn, aims to produce buildings whose rendered appearance faithfully reflects their textual specifications. We introduce a **Visual Consistency Reward** that assesses alignment in terms of facade details, style, and material coherence with the input text, guiding BuildingGen to generate 3D buildings that are both semantically faithful and visually coherent.

By employing programs as editable proxies and designing the reward to enhance generalization capabilities, our system further enables fine-grained control over city elements. Users can directly modify blocks or buildings through natural language commands, including changes to style, structure, and spatial distribution, without relying on external tools or plugins.

Overall, CityGenAgent combines the broad world knowledge of LLMs with domain-specific fine-tuning, yielding a framework that is capable of generating the realistic and editable city scene.

In summary, our main contributions are:

- We propose programs specifically for 3D city generation, Block Program and Building Program. This approach decomposes city into blocks, buildings, and building components, enabling flexible control and executed.

- We introduce CityGenAgent, comprising two core modules: BlockGen and BuildingGen. By introducing Spatial Alignment Reward and Visual Consistency Reward, we enhance the model's spatial reasoning and ensure coherent visual fidelity.

- Experimental results demonstrate that CityGenAgent is capable of accurately following user instructions to generate high-quality 3D cities. Furthermore, the system supports users to interactive manipulate the block and building by natural language.

## 2 RELATED WORK

**Rendering and Diffusion-based Scene generation.** Scene generation is typically addressed through rendering-based and diffusion-based methods. Neural rendering-based approaches (Chen et al., 2023; Lin et al., 2023; Xie et al., 2024; Shen et al., 2022) use implicit representations of 3D scenes and apply volumetric rendering to neural fields. For instance, CityDreamer (Xie et al., 2024) segments the urban environment into buildings and backgrounds, employing distinct neural field types. While these methods achieve impressive visual quality, their lack of 3D geometric fidelity and user control restricts their applicability in downstream tasks. Some research has increasingly explored the use of diffusion-based methods to generate layouts or scenes (Inoue et al., 2023; Wu et al., 2024; Yu et al., 2024; Ren et al., 2024; Bian et al., 2024). DynamicCity Bian et al. (2024) employs the voxel-based representation for large-scale city generation. However, this approach inherently lacks fine-grained geometric details and fails to capture the high-fidelity structural intricacies characteristic of real-world. CityGen (Deng et al., 2025) introduces an end-to-end framework capable

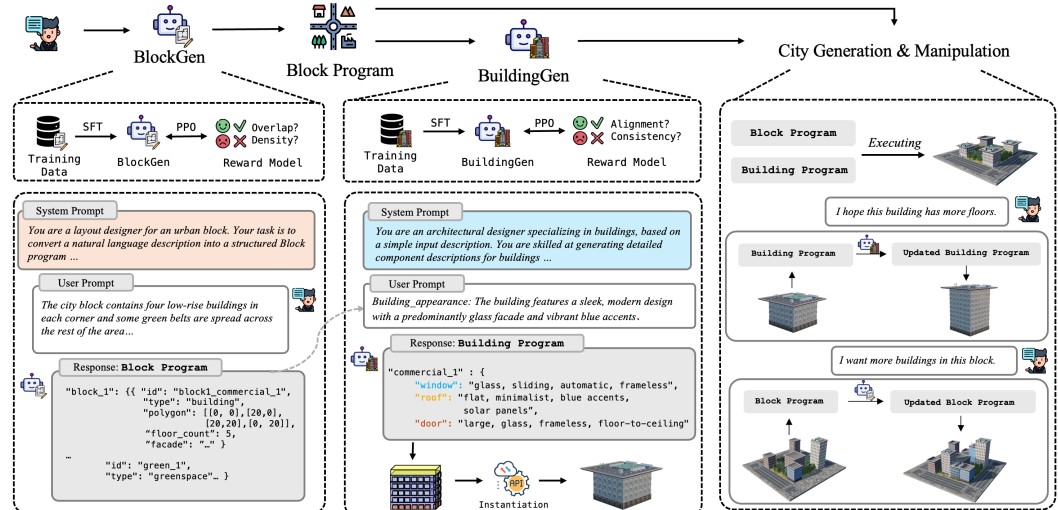

Figure 1: **Overview of the proposed CityGenAgent framework.** The pipeline decomposes city generation into two stages. **BlockGen** converts user prompt into structured Block Program that defines spatial layouts of urban elements. **BuildingGen** refines each block by producing Building Program that captures architectural attributes. Block Program and Building Program are then executed into 3D city instances, which can be interactively manipulated via natural language refinement.

of generating diverse city layouts using Stable Diffusion but demonstrates limited extensibility in accommodating conditional inputs for subsequent editing and refinement operations.

**Procedure-based Scene Generation.** Traditional approaches (Parish & Müller, 2001; Kelly & McCabe, 2007) have established rule-based approaches to generate road networks and buildings, which demand extensive manual modeling and substantial labor costs. Recently, methods like Infinigen (Lee et al., 2024) and Infinigen Indoors (Raistrick et al., 2024) introduce comprehensive procedural systems for generating natural landscapes and indoor scenes through stochastic or constrained mathematical algorithms, yielding highly diverse and photorealistic outcomes. With the development of LLMs (Ouyang et al., 2022), researchers have made attempts to sythesis scenes conditioned on user input, including general scenes (Zhang et al., 2025; Gao et al., 2024; Zhou et al., 2024; Sun et al., 2025a; Liu et al., 2025) and indoor scenes (Feng et al., 2023; Sun et al., 2025b; Fu et al., 2024). As for urban environments, CityCraft (Deng et al., 2024) and UrbanWorld (Shang et al., 2024) leverage the natural language capabilities of LLMs for urban layout generation. However, these methods lack explicit structural definitions of urban patterns and exhibit strong reliance on pre-existing priors, which consequently constrains their diversity and scalability. CityX (Zhang et al., 2024) proposes a multi-agent collaborative framework but it relies heavily on contextual plugin coordination, making it difficult to guarantee the correct execution of the overall pipeline.

## 3 METHOD

### 3.1 OVERVIEW

Given an input description of a city block $I$, our goal is to generate a visually coherent and semantically consistent 3D city block $H$. Figure 1 provides an overview of our framework, **CityGenAgent**, which integrates two core components: BlockGen (left), which converts user prompt into Block Program that specifies spatial layouts of buildings and block elements, and BuildingGen (middle), which further refines each block by generating Building Program that captures building appearance details. BlockGen and BuildingGen are finetuned based on LLMs in two stages: SFT on instruction–program pairs, followed by Proximal Policy Optimization (PPO) to enhance spatial reasoning and visual consistency. Both programs serve as editable intermediates that can be executed (right) to assemble the final 3D city, and more importantly, enable manipulation of the generated city. More detailsof BlockGen and BuildingGen are provided in Sections 3.2 and 3.3.

## 3.2 BLOCKGEN

**Goal and interface.** Given a description $I$ of a city block, **BlockGen** outputs Block Program $P_{block}$ that parameterizes the block layout, including the placement and attributes of buildings and other elements like greenspaces. BlockGen is trained to map user instructions into Block Program and we refine it through SFT and PPO to enhance the spatial reasoning (details in the following sections).

**Block Program.** A Block Program $P_{block}$ encodes the block layout as an ordered list of elements $P_{block} = \langle b_1, \ldots, b_n \rangle$, where each element $b$ contains the following fields. The first three fields are required, while the last two are optional and apply only to buildings. An example Block Program is provided in Appendix C.

- `id` (string, required): A unique identifier for the element.
- `type` (string, required): The usage category of the element, such as `"residential"`.
- `polygon` (list of $[x, y]$, required): A simple (non-self-intersecting) footprint represented as a counter-clockwise ordered list of 2D vertices in block coordinates (meters).
- `floor_count` (integer $\geq 1$, optional): The number of floors for a building.
- `facade` (string, optional): A natural-language description of the building's facade appearance.

### 3.2.1 BLOCKGEN SUPERVISED FINE-TUNING (BLOCK-SFT)

The base LLMs often violate the basic schema (e.g., missing fields, ill-formed polygons) and weakly adheres to city layout priors. SFT addresses these issues by teaching the exact schema and output conventions. To achieve this, we synthesize a paired dataset and each pair contains an input prompt and its target Block Program. The raw pairs are post-processed to remove low-quality samples, shown in Appendix D. In this process, BlockGen is trained to generate the valid Block Program, capturing basic spatial relationships among block elements and aligning them with user instructions.

### 3.2.2 SPATIAL ALIGNMENT REWARD PREFERENCE OPTIMIZATION (BLOCK-PPO)

Simple SFT on our limited synthetic dataset reliably teaches BlockGen to produce well-formed block programs, but does not yield robust spatial reasoning or generalization to complex, unseen scenarios. We therefore design specific rewards and adopt RL to enhance the spatial reasoning of our BlockGen. Concretely, we define **Spatial Alignment Reward** that scores each generated Block Program from two complementary perspectives: **Semantic Consistency**, which measures its consistency to the input descriptions $I$, e.g., correct types and relative placements, and **Spatial Structural Consistency**, which encourages physically plausible layouts, e.g., non-overlapping footprint. This reward evaluation allows the model to move beyond SFT's format alignment and learn policies that generalize to more complex and even out-of-distribution layouts.

**Semantic Consistency Evaluation.** The evaluation is to quantify how well a predicted Block Program semantically aligns with the user instruction $I$. Measuring semantic alignment is challenging since text provides high-level instructions while the Block Program specifies low-level geometry. To bridge this gap, we render the program as a 2D image and use a Vision-Language Model (VLM) to assess semantic alignment and global plausibility. Specifically, we rasterize polygon footprints with type-specific fills and boundaries, then feed the rendered image and text into GPT-4o (Achiam et al., 2023) with a standardized prompt (see Appendix A.1) to obtain two scalar scores:

- Semantic Alignment $S_{align} \in [0, 10]$, assessing faithfulness of the layout to the text.
- Global Plausibility $S_{plau} \in [0, 10]$, assessing whether the arrangement is physically plausible.

These scores are then used to calculate our final Spatial Alignment Reward score.

**Spatial Structural Consistency Evaluation.** Beyond semantic alignment and global plausibility, good urban layouts should (i) avoid overlap of elements and (ii) maintain a reasonable built-area coverage. For the given block program, we therefore introduce two simple but broadly applicable priors: Geometric Overlap and Footprint Density, measuring the interpenetration and building-area coverage of the block program to form our spatial objective. We next detail how each score is defined and computed.

- Geometric Overlap ($S_{\text{overlap}}$):

  Given a Block Program $P_{\text{block}} = \langle b_1, \ldots, b_n \rangle$, each building $b_i$ stores fields such as `id`, `type`, and `polygon`. We use only the `polygon` for geometric overlap. Let $L$ be the area of the block region, $A(\cdot)$ the area operator and $R_i$ denote the area of the $i$-th `polygon`. For each $b_i$, let

  $$\text{poly}(b_i) = \langle (x_{i,1}, y_{i,1}), \ldots, (x_{i,m_i}, y_{i,m_i}) \rangle. \tag{1}$$

  be its simple (non-self-intersecting) polygon. The axis-aligned bounding box (AABB) of $b_i$ is defined as

  $$R_i = [x_i^{\min}, x_i^{\max}] \times [y_i^{\min}, y_i^{\max}], \tag{2}$$

  $$x_i^{\min} = \min_k x_{i,k}, \; x_i^{\max} = \max_k x_{i,k}, \; y_i^{\min} = \min_k y_{i,k}, \; y_i^{\max} = \max_k y_{i,k}. \tag{3}$$

  We then define the geometric overlap percentage $O$ as follows:

  $$O = \frac{\displaystyle\sum_{i<j} A\big(R_i \cap R_j\big)}{A(L)}. \tag{4}$$

  i.e., the total overlapping area among all pairs of building AABBs. We use AABBs rather than exact polygon intersections because they are fast, numerically stable under small vertex jitter, and conservative (upper-bound true polygon overlap), providing a simple but robust penalty signal during training.

  Overlap percentage $O$ is normalized to 0–10 scale to obtain the $S_{\text{overlap}}$:

  $$S_{\text{overlap}} = 10 \times (1 - O). \tag{5}$$

  The highest score is assigned when there is no overlap, and the score decreases as the degree of overlap increases.

- Footprint Density ($S_{\text{density}}$):

  To complement $S_{\text{overlap}}$, we assess built-area coverage using the same AABBs $\{R_i\}$. Let $[D_{\min}, D_{\max}]$ the pre-given desired density band. We first compute the overall density $D$ as follows:

  $$D = \frac{\sum_{i=1}^N A(R_i)}{A(L)}. \tag{6}$$

  We then define density score $S_{\text{density}}$ to promote layouts whose coverage lies within the desired band while penalizing under- or over-development:

  $$S_{\text{density}} = \begin{cases} 10, & \text{if } D_{\min} \le D \le D_{\max}, \\[2mm] 10 \times \dfrac{D}{D_{\min}}, & \text{if } D < D_{\min}, \\[2mm] 10 \times \dfrac{D_{\max}}{D}, & \text{if } D > D_{\max}. \end{cases} \tag{7}$$

  Densities within the target range $[D_{\min}, D_{\max}]$ receive higher scores, whereas overly dense or overly sparse configurations result in lower scores. We set $D_{\min}$ to 0.5, while $D_{\max}$ to 0.8 to ensure the efficiency and viability of the block.

**Spatial Alignment Reward.** We define the final Spatial Alignment Reward as the mean of four reward scores: semantic alignment ($S_{\text{align}}$), global plausibility ($S_{\text{plau}}$), geometric overlap ($S_{\text{overlap}}$), and footprint density ($S_{\text{density}}$):

$$S_{\text{spatial}} = \frac{1}{|\mathbb{S}|} \sum_{S_i \in \mathbb{S}} S_i, \qquad \mathbb{S} = \{S_{\text{align}}, S_{\text{plau}}, S_{\text{overlap}}, S_{\text{density}}\}. \tag{8}$$

This simple averaging keeps contributions balanced without extra hyperparameters; a weighted variant can be used if different priorities are desired.

**Preference Optimization.** We use PPO (Schulman et al., 2017), a reward-based algorithm, to enhance the spatial reasoning of BlockGen. Following the common practice (Ouyang et al., 2022), we construct the preference pairs for training the reward model to predict a scalar score that reflects the target preference. Then we use the output of the reward model as the reward signal to supervise the policy model, enhancing the spatial reasoning.

### 3.3 BUILDINGGEN

**Goal and interface.** Given the building facade description in `facade` key of Block Program, BuildingGen is trained to map the description to Building Program $P_{\text{building}}$, which decomposes the building into distinct components and provides detailed feature for each component. We fine-tune BuildingGen on LLMs via SFT and PPO for semantic alignment and visual consistency.

**Building Program.** A Building Program $P_{\text{building}}$ encodes the appearance of a building into the description of its components $P_{\text{building}} = \langle c_1, \ldots, c_n \rangle$, where each component $c$ has two required fields, defined as follows, with an example shown in Appendix C.

- `type` (string): A category to describe the usage of the component, such as `"door"`.
- `description`: A natural language, consisting of several phrases, describing the component's color, style, material, and other decorative details, such as `"large, blue, frameless"`.

#### 3.3.1 BUILDINGGEN SUPERVISED FINETUNING (BUILDING-SFT)

Similar to BlockGen, we use SFT to warm the LLMs. To achieve this, we construct a synthetic dataset of paired samples for SFT. Each pair consists of a building appearance description and its corresponding component-based Block Program. This step enables BuildingGen to acquire basic format mapping and semantic understanding capabilities, which are essential for subsequent capability enhancement and generalization.

#### 3.3.2 VISUAL CONSISTENCY REWARD PREFERENCE OPTIMIZATION (BUILDING-PPO)

A modality gap remains after Building-SFT: a text-only model lacks visual grounding, so program text may not match the rendered appearance. We address this with **Visual Consistency Reward**: execute the generated Building Program to get the renderings, use a VLM to score the visual consistency, train the reward model on these scores, and optimize the policy model using PPO with the reward signal. This closes the gap and steers the model toward programs that render faithfully to the prompt.

**Visual Consistency Reward.** We develop the visual criteria to assess Visual Consistency Reward score of the rendered buildings based on these key aspects.

- Text Alignment: Examines the alignment between the visual result and the input prompt.
- Color Coherence: Assesses whether the color scheme across the building is harmonious.
- Style Consistency: Evaluates the consistency of architectural styles among components.
- Material Coherence: Focuses on the compatibility of materials used throughout the facade.

**Preference Optimization.** Following the workflow in BlockGen, we use the defined visual criteria to construct the dataset to train the reward model and policy model. We carefully designed the evaluation prompt based on these four criteria, as shown in Appendix A.1. Further training details are provided in Appendix E.

### 3.4 PROGRAM EXECUTION AND ASSET ASSEMBLY

With the Block Program and Building Program, our executor generates 3D city scenes in two stages. **Asset Preparation**: parse the Block Program to extract footprints and floor counts, build base meshes, then use Building Program component descriptions to retrieve or synthesize assets (doors, windows, roofs). The component descriptions for windows or doors from the Building Program are used to perform semantic retrieval from an architectural asset library. To address the limitations of the fixed asset library, we further experimented with using the component description as prompts for Text-to-3D models (e.g., Hunyuan3D (Zhao et al., 2025)), thereby enabling dynamic expansion of our component database. The geometry attributes from the Block Program are used to compute the parameters required for asset placement. For each edge of the polygon, we calculate its length, direction, and outward normal vector to compute the transform matrix of each component along that edge. **Asset Assembly**: instantiate these assets in a fixed order and place them onto the structure via

**SGAM**

**Infinicity**

**CityDreamer**

**CityCraft**

**CityGenAgent (Ours)**

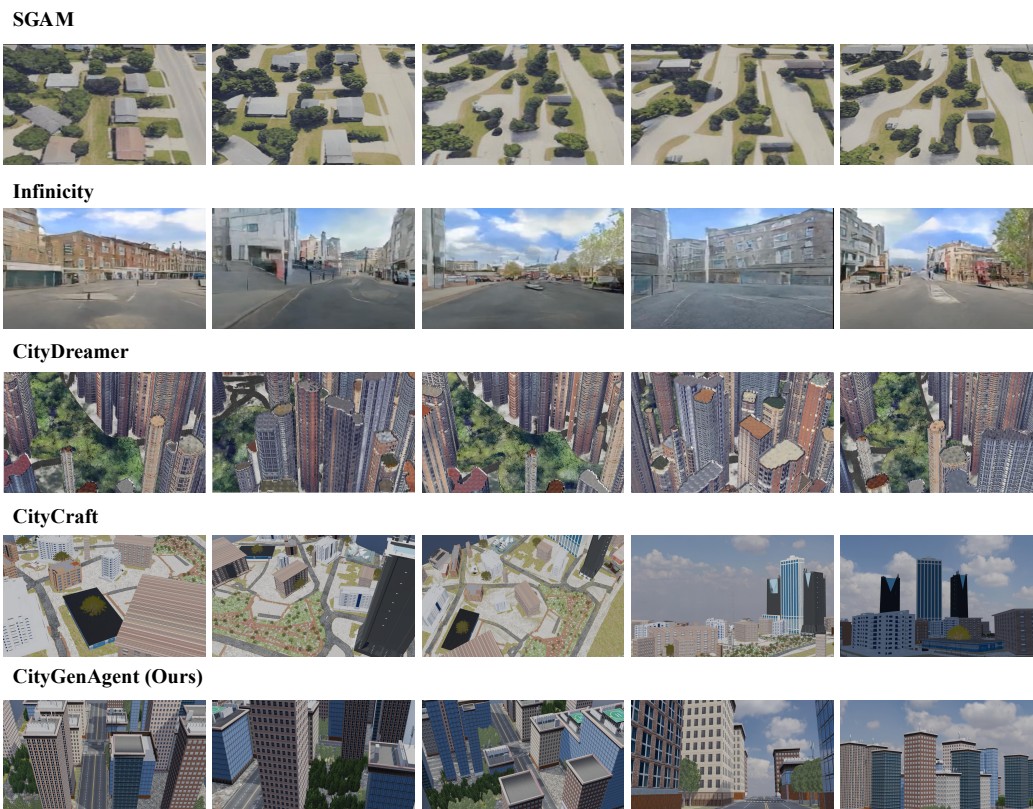

Figure 2: **Comparison Results of City Generation.**

rotation, translation, and scaling to form complete buildings. At this stage, based on the transformation parameters and assets prepared during the Asset Preparation phase, we implement an executor that performs graphics operations such as rotation, translation, scaling and placement to accurately position each component at its designated location, assembling the final reasonable 3D buildings. Meanwhile, other scene elements, such as roads, trees, and streetlights, are generated based on the spatial positions specified in the Block Program. Acting as an intermediary, the executor translates program instructions into commands compatible with standard graphics software such as Blender and Unreal Engine. This enables the automated and scalable generation of complex 3D city models from text, ensuring seamless integration into professional visualization and development workflows.

### 3.5 INTERACTIVE MANIPULATION VIA LANGUAGE

Leveraging the effective representations of Block Program and Building Program, along with the model's generalization capability acquired through RL, our framework enables users to manipulate individual blocks or buildings via natural language commands. Given the current Block Program or Building Program, the user can provide instructions to the corresponding module, BlockGen or BuildingGen, which then updates the program accordingly to follow the desired changes. For example, BlockGen can modify block density or adjust building heights, while BuildingGen can alter architectural details such as the style of windows and doors.

## 4 EXPERIMENTS

### 4.1 EXPERIMENTAL DETAILS

**Dataset.** We design prompts carefully and use GPT-4o to synthesize data for constructing the training dataset. All generated samples are validated to ensure the quality. For evaluation, we gather 50 city block descriptions and 50 manipulation prompts, subsequently input into CityGenAgent to

generate 3D scenes for quantitative and qualitative comparisons. The details for constructing dataset is provided in the Appendix D.

**Model Training.** In our framework, BlockGen and BuildingGen are both finetuned from QWen3-8B (Yang et al., 2025). Training details are provided in Appendix E.

**Metrics.** We evaluate the rendered 3D city scenes using *Text Alignment* and *Visual Consistency* from GPT and a user study. The CLIP score (Radford et al., 2021) is also calculated in the evaluation. We use two indicators for **geometric quality** of 3D mesh: ROS for edge orthogonality and OTR for tessellation efficiency. Please refer to Appendix B for details. For BLockGen, we evaluate three metrics: *Collision Rate* (Collision), *Positional Coherency* (Pos.), and *Physics-based Semantic Alignment* (PSA). Collision is defined as the ratio of total pairwise AABB overlap area to the block area, while Pos. and PSA follow LayoutVLM (Sun et al., 2025b). For BuildingGen, we use GPT-4o to assess *Text Alignment* and *Visual Consistency* of individual buildings. To quantitatively assess whether the generated program complies with the specification, we adopt a metric termed *Format Accuracy*. This evaluation encompasses three key criteria: (i) the validity of the output as a parsable JSON structure, (ii) the geometric correctness of polygon definitions, and (iii) the completeness of required fields within the generated programs.

## 4.2 COMPARISON WITH EXISTING METHODS

We compare the generated city scenes with existing approaches, with a summary of these methods provided in Appendix H. For methods that output images, we assess rendering results using established metrics. We conduct a comparative analysis of geometric quality indicators across our method, Hunyuan3D (Zhao et al., 2025) and CityCraft (Deng et al., 2024).

Table 1: **Quantitative Comparison on Text Alignment and Visual Consistency.**

| Method | Text Alignment ↑ | | | Visual Consisitency ↑ | | Geometric Quality | |
|---|---|---|---|---|---|---|---|
| | CLIP | GPT | User | GPT | User | ROS↑ | OTR↓ |
| SGAM (Shen et al., 2022) | - | - | - | 5.1 | 4.2 | - | - |
| Infinicity (Lin et al., 2023) | - | - | - | 4.0 | 2.9 | - | - |
| CityDreamer (Xie et al., 2024) | - | - | - | 6.0 | 4.1 | - | - |
| CityCraft (Deng et al., 2024) | - | - | - | 6.1 | 5.1 | 0.309 | 192.301 |
| Hunyuan3D (Zhao et al., 2025) | 0.272 | 5.2 | 3.9 | 6.5 | 5.5 | 0.182 | 6999.983 |
| CityGenAgent(Ours) | **0.286** | **6.6** | **6.1** | **6.7** | **5.8** | **0.357** | **177.970** |

Table 2: **Quantitative Comparison of Different Language Models in City Generation.**

| Method | Format Accuracy | Collision | Pos. | PSA |
|---|---|---|---|---|
| GPT-4o (Achiam et al., 2023) | 70% | 6.67% | 78.45 | 85.10 |
| Qwen2.5-7B (Team et al., 2024) | 70% | 37.99% | 67.60 | 61.25 |
| Qwen3-8B (Yang et al., 2025) | 83% | 23.97% | 76.70 | 75.60 |
| CityGenAgent w/o RL | 98% | 5.59% | 80.17 | 84.02 |
| CityGenAgent | **98%** | **4.89%** | **85.33** | **87.90** |

**Quantitative Comparison.** As shown in Table 1, our method outperforms existing city generation approaches across *Text Alignment* and *Visual Consistency*. While CityCraft supports text input, its implementation is not open-sourced. For geometric quality, CityGenAgent demonstrates better rectilinearity (ROS) and an order-of-magnitude gain in tessellation efficiency (OTR), reducing over-tessellation by approximately $40\times$, compared to Hunyuan3D. This advantage stems from our procedural generation strategy, which ensures structural regularity and more efficient mesh distribution. From Table 2, the results indicate that CityGenAgent achieves superior overall performance compared to both large language model baselines and ablation settings. It maintains high format accuracy and low collision rates, which is critical for generating structurally sound urban layouts. The

RL module enhances spatial reasoning while preserving format accuracy, leading to improved performance in both collision avoidance and semantic consistency. This demonstrates the effectiveness of the reward design in the training process.

**Efficiency Evaluation.** To illustrate the efficiency of CityGenAgent , we provide the time required of our method compared with Hunyuan3D (Zhao et al., 2025), CityCraft (Deng et al., 2024) and human craft. The experiments are conducted on a server equipped with an NVIDIA H100 NVL GPU for inference and rendering tasks, with a typical memory footprint of approximately 64 GB during execution. Additionally, we consult PCG experts to determine the time needed to construct the same scale scene. The comparison results are shown in Table 3. From the results, we can observe that CityGenAgent is faster than other methods in generating the scene from a description for the block, maintaining a moderate memory. This combination of reduced computational time and memory usage underscores CityGenAgent's efficiency and its suitability for real-time generation and deployment in resource-constrained environments.

**Qualitative Comparison.** As illustrated in Figure 2, the qualitative results highlight the superiority of CityGenAgent. In comparison, InfiniCity, SGAM, and CityDreamer all suffer from low clarity and a lack of urban details. Although CityCraft demonstrates coherent 3D structures but the scale and harmony between buildings lack consistency and do not conform to real-world rules. As illustrated in Figure 3, Hunyuan3D's renderings are constrained to a cartoon-like style and exhibit limited scalabil-

Table 3: **Efficiency Evaluation of City Generation Methods for Per Block.**

| Method | Inference Time | VRAM |
|---|---|---|
| human | 60min | - |
| Hunyuan3D | 3 min | 16GB |
| CityCraft | 1 min | - |
| CityGenAgent | **0.75 min** | **8GB** |

ity. Mesh and wireframe visualizations further reveal that it often produces irregular and overly dense tessellations, resulting in redundant geometry. In contrast, our method preserves generative flexibility while producing clean, planar surfaces with significantly improved mesh regularity and structural clarity, leading to superior visual and geometric quality. Additional results of CityGenAgent are presented in Figure 4 and Appendix F.

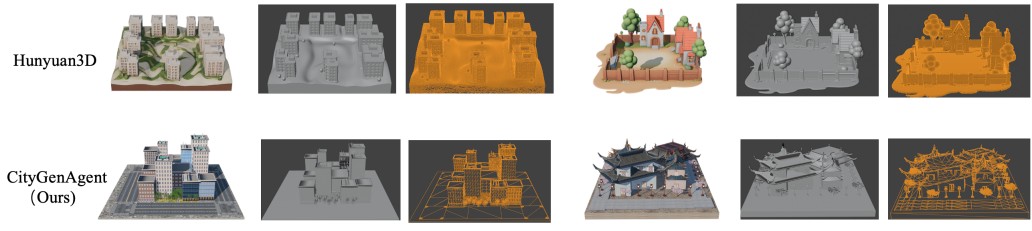

Hunyuan3D

CityGenAgent
(Ours)

*"The urban block contains 12 residential buildings and 2 greenspaces. "*

*"An ancient style block with four houses neatly arranged."*

Figure 3: **Qualitative Comparison with Hunyuan3D.** We present the prompts, rendered images, mesh visualization, and wireframe visualization for each scene.

### 4.3 MANIPULATION

The city block produced by our framework can be further manipulated using natural language, as illustrated in Figure 5. Since Block Program and Building Program reflect the structural layout of the scene and the composition of building components, modifying program parameters leads to precise adjustments in the scene while preserving plausibility and aesthetic quality. For example, building floor counts can be adjusted using fuzzy or numeric language, architectural style by modifying component descriptions, and building density and layout through spatial manipulations. Existing open-source text-to-3D scene generation methods, such as CityCraft and Hunyuan3D, do not support manipulation of city scenes via natural language, leaving no suitable baselines for comparison. We further provide additional qualitative and quantitative evaluations in Appendix G.

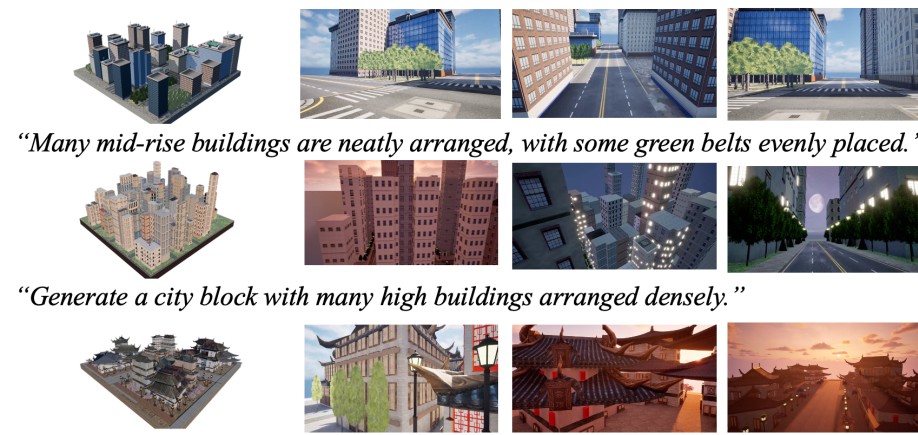

Figure 4: **Visual Results Generated by CityGenAgent.** Results are shown across diverse conditions, including daytime, dusk, nighttime, and ancient style.

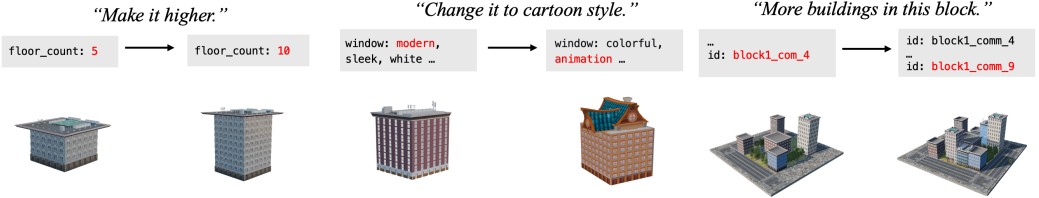

Figure 5: **Scene Manipulation with Natural Language.**

## 4.4 ABLATION STUDY

We examine the effects of our reward design performance by evaluating the system when each part is removed. Results show that incorporating the reward consistently improves model performance across all metrics. Detailed results and analysis are provided in the Appendix I. We compare different RL methods, as shown in Table 4. We can observe that applying PPO and DPO after SFT provide consistent improvements across all metrics, which further validates the effectiveness of our reward design.

Table 4: **Ablation Study**. We evaluate different RL methods for BlockGen and BuildingGen.

<table>
<tr><td colspan="4">(a) BlockGen</td><td colspan="3">(b) BuildingGen</td></tr>
<tr><td>**Method**</td><td>**Collision** ↓</td><td>**Pos.** ↑</td><td>**PSA** ↑</td><td>**Method**</td><td>**Text Alignment** ↑</td><td>**Consistency** ↑</td></tr>
<tr><td>Base Model</td><td>23.97%</td><td>76.70</td><td>75.60</td><td>Base Model</td><td>5.5</td><td>5.7</td></tr>
<tr><td>Base Model + SFT</td><td>5.59%</td><td>80.17</td><td>84.02</td><td>Base Model + SFT</td><td>6.8</td><td>8.7</td></tr>
<tr><td>Base Model + DPO</td><td>5.19%</td><td>81.13</td><td>85.03</td><td>Base Model + DPO</td><td>7.0</td><td>8.1</td></tr>
<tr><td>Base Model + PPO</td><td>**4.89%**</td><td>**85.33**</td><td>**87.90**</td><td>Base Model + PPO</td><td>**7.5**</td><td>**8.9**</td></tr>
</table>

## 5 CONCLUSION

We present CityGenAgent, a natural language-driven framework for hierarchical procedural generation of 3D cities. By introducing Block Program and Building Program, we enable precise control over the spatial, architectural, and stylistic attributes of city elements. BlockGen and BuildingGen are trained through SFT and RL with our designed reward to ensure spatial reasoning and visual consistency. Unlike prior methods that rely on fixed assets or coarse control, our approach supports fine-grained manipulation through natural language. Our work lays the foundation for scalable and user-friendly 3D city modeling, and opens new possibilities for city design and interactive content creation. Further discussion of limitations is provided in Appendix K.

ETHICS STATEMENT

This work does not involve human subjects, sensitive personal data, or experiments that could raise privacy or security concerns. Our methodology does not introduce harmful insights or applications, and we have taken care to avoid bias or discrimination in both data and model design.

REPRODUCIBILITY STATEMENT

We provide detailed descriptions of our proposed frameworks in Section 3, including all architectural components, their interactions, and the full pipeline for transforming natural language inputs into 3D outputs. Section 4.1 outlines the experimental setup, including datasets, model configurations, and evaluation metrics.

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

APPENDIX

# A  IMPLEMENTATION DETAILS

## A.1  PROMPTS

In this section, we provide prompts used for training and testing the model.

**BlockGen-SFT Data Generation.** In order to enable BlockGen to output the corresponding program in the SFT stage, we use the following prompts to generate Block Program.

```
You are a layout parser for urban block descriptions.
Your task is to convert a natural language description of an
urban block into a structured JSON layout.  Each building or
green space must be assigned:
- a unique "id",
- a "type" (such as "resident building" ,"school", "library",
"commerical", building", "office", "greenspace"),
- a "polygon" represented as a list of [x, y] points (for
example, x, y are integers within [0, 100]),
- and for buildings, there is a key "floor count" should be
a digital number and a key "facade" to describe the general
appearance of the building.

Make sure the polygons do not overlap and collectively cover
the area proportions mentioned in the description.  Keep the
shapes simple (e.g., rectangles or L-shapes), and follow the
stated quantity and proportions as accurately as possible.

Your output must follow **this exact JSON format**:
[
{
    "description": "<repeat the input description here>",
    "layout":
    {
    "buildings": [
    {
        "id": "res_1",
        "type": "resident building",
        "polygon": [[x1, y1], [x2, y2], [x3, y3], [x4, y4]],
        "floor_count": 16,
        "facade": "glass with wooden accents"

    },
    ...
    ]
    "greenspaces": [
    {
        "id": "green_1",
        "type": "greenspace",
        "polygon": [[x1, y1], [x2, y2], [x3, y3], [x4, y4]]
    },
    ...
    ]
    }
}
```

```
]
Prompt:
{Text description}
```

**BuildingGen-SFT Data Generation.** In order to enable BuildingGen to output the corresponding program in the SFT stage, we use the following prompts to instruct GPT-4o to generate Block Program.

```
You are an architectural designer specializing in buildings.
Based on a single frontal image of a building facade,
produce structured descriptions suitable for 3D modeling or
procedural generation.

Obey the following rules:  - Input:  one frontal image of the
building facade.
- Output must strictly be valid JSON, with no additional text
or explanations.
- The JSON is an array containing one object with exactly two
top-level keys:  "facade" and "output".
- "facade":  a single short sentence summarizing the overall
facade appearance (mention color, style, material, and key
characteristics).  Keep it concise and continuous, not a list
of words.
- "output":  an object with exactly these
keys | "window", "door", "roof".  Each value
must be a concise, comma-separated phrase
(style/material/structural/ornamental/color descriptors; no
sentences, no subkeys, no nesting; no trailing commas).

Here is the output format you must follow:
[
  {
    "facade": "The facade features a modernist design with
      light gray concrete and large glass panels.",
    "output": {
      "window": "rectangular modules, aluminum frames, clear
        glass, repetitive grid, slim mullions",
      "door": "single-leaf, glass panel, metal frame, flush
        alignment, minimal handle",
      "roof": "flat slab, parapet edge, concealed drainage,
        concrete, clean silhouette"
    }
  }
]
```

**Semantic Consistency Evaluation.** For Block-PPO, we calculate the semantic consistency score of samples to construct the positive and negative sample pairs. To evaluate the semantic consistency of the Block Program sample, we use the following prompt as input to GPT-4o.

```
You are an urban planning expert and are asked to analyze
and score the city block layout image based on the text
description.  Please strictly follow the following criteria
and output the result in a structured JSON format.
```

```
Blue represents buildings, and green represents green space.

Please refer to the following criteria:
1.  **Semantic Alignment**:  Does the layout conform to
the text description, especially regarding quantity,
distribution, and orientation?
2.  **Global Plasusibility**:  Does the overall layout
exhibit physical feasibility and structural coherence?
Are the spatial arrangements consistent with real-world
constraints such as accessibility and functional
organization?

Please:
- Read and analyze the image file;
- Assign an integer score (0 to 10) to each dimension;
- Save the results in the following JSON format.

The output format is as follows:

{
    "image_1":
    {
        "semantic_alignment": 6,
        "global plasusibility": 8
    }
}

Text description:
{text_description}
Block_layout:
{block_layout_img}
```

**Visual Consistency Evaluation.** For Building-PPO, we evaluate the visual consistency score of the rendered result to obtain the preference data pairs to train the reward model. We use the following prompt to evaluate the visual results of the executed Building Program.

```
You are given a facade view image of a building.  Please
evaluate the image from the following perspectives:
- Text Alignment:  Examine the degree of correspondence
between the completed facade design and the text description
provided.  Consider whether the overall style, material,
color scheme, and structural features match the description.
- Color Coherence:  Assess whether the color scheme across
the facade is harmonious and visually consistent.
- Style Consistency:  Evaluate the consistency of
architectural styles among components such as windows, doors,
and roofs.
- Material Coherence:  Focus on the compatibility and
uniformity of materials used throughout the facade.
For each aspect, provide:
- A score from 1 to 10 (1 = very poor, 10 = excellent)
- A brief explanation for your rating
**Output format:**

Text Alignment: [score]/10 | [explanation]
Color Coherence: [score]/10 | [explanation]
```

```
Style Consistency: [score]/10 | [explanation]
Material Coherence: [score]/10 | [explanation]

Text description:
{Text_description}
Input image:
{Input_img}
```

## B  EVALUATION

**GPT-based Evaluation.** To evaluate the quality of the generated 3D cities, we instruct GPT-4o to give the score from the specific aspects. The prompt is shown as follows.

```
You are given a top-down view image and a text description
of a city block.  Please evaluate the scene based on the
following aspects:

Consistency:
Color Cohesion:  Are the colors of windows, doors, and roofs
across different buildings in the block harmonious?
Style Consistency:  Do the buildings generally belong to a
similar architectural style or era?
Authenticity:  Does the block conform to the morphology and
layout of real-world urban environments?
Quality:  Is the image clear and high-fidelity?
Text Alignment:  Does the appearance match the given input
description?

For each aspect, provide a score from 1 to 10:
0-3:  Severe Dissonance (clashing, chaotic, lacking cohesion)
4-6:  Neutral/Mixed (partially harmonious but with noticeable
inconsistencies)
7-8:  Harmonious (cohesive and intentional, with minor
acceptable variations)
9-10:  Highly Harmonious/Uniform (excellent cohesion; tells a
clear, unified design story)

A brief explanation for your rating.
Output format:
Consistency:  [score]/10 | [explanation]
Text Alignment:  [score]/10 | [explanation]

Text description:
{Text_description}
Input image:
{Input_img}
```

**User Study Details.** In our experiments, we employ manual evaluation to assess the generated results. We recruited 30 volunteers to participate in the scoring process. An example of the evaluation interface is shown in Figure 6. To ensure fairness and objectivity, all evaluation images were anonymized to eliminate potential bias. This procedure helped maintain the integrity and reliability of the evaluation process.

**Geometric Quality.** To assess geometric quality of our method, we use the architectural regularity and efficiency indicators. (i) **Rectilinearity/Orthogonality Score (ROS)** measures the proportion

Figure 6: **Template Questionnaire Used in Participant Studies.**

of edge directions aligned with two dominant orthogonal axes, reflecting facade alignment and structural order. Higher ROS indicates better orthogonality. (ii) **Over-tessellation Ratio (OTR)** compares actual triangle density to curvature-based demand, where lower values indicate more efficient tessellation without unnecessary mesh complexity.

## C    EXAMPLES OF BLOCK PROGRAM AND BUILDING PROGRAM

**Block Program.** As follows, we provide an example for Block Program.

```
{
    "id": "mixed_1",
    "type": "mixed-use building",
    "polygon": [[0, 0], [22, 0], [22, 22], [0, 22]],
    "floor_count": 12,
    "facade": "modern glass and steel with terracotta accents"
},
{
    "id": "mixed_2",
    "type": "mixed-use building",
    "polygon": [[25, 0], [47, 0], [47, 22], [25, 22]],
    "floor_count": 10,
    "facade": "concrete with greenery on the upper floors"
},
...
{
    "id": "park_1",
    "type": "greenspace",
    "polygon": [[36, 50], [55, 50], [55, 67], [36, 67]]
},
{
    "id": "park_2",
    "type": "greenspace",
    "polygon": [[36, 71], [55, 71], [55, 89], [36, 89]]
},
...
```

**Building Program.** As follows, we provide an example for Building Program.

```
{
   "window": "expansive, glass, modern, blue-tinted",
   "door": "sleek, modern, glass, automatic",
   "roof": "flat, sleek, modern, weather-resistant"
}
```

## D  DATASET CONSTRUCTING

**Training Dataset For BlockGen.** For BlockGen, we curate 1,000 valid samples for the SFT stage. The raw pairs are post-processed to remove invalid or low-quality samples. Concretely, we (i) verify each `polygon` is a closed simple loop vertices and counter-clockwise ordering; (ii) reject pairs with overlapping between polygons (edge/vertex touching is allowed); and (iii) enforce *appropriate density* so blocks are neither empty nor overfilled. To construct preference pairs for RL training, we start with 1,000 text prompts describing blocks. For each prompt, GPT generates five candidate outputs. We then score these outputs using our reward rule on a 0–10 scale. From each set of five, we select the highest-scoring and lowest-scoring samples, provided their reward difference is at least 5. These pairs are labeled as "chosen" and "rejected," resulting in 1,000 preference pairs for training the PPO model.

**Training Dataset For BuildingGen.** For BuildingGen, since there are few building datasets suitable for this task, we construct a paired dataset of 5,000 examples. Each pair consists of a natural language building description and its corresponding procedural program. To synthesize this dataset, we collected 5,000 frontal building images from Google Maps and designed specialized prompts for GPT-4o to generate both holistic descriptions of the building facade and component-level descriptions based on our predefined architectural categories such as door, window, and roof. For training the reward model, we gather 5000 diverse prompts and generate 5 samples for each using GPT-4o.

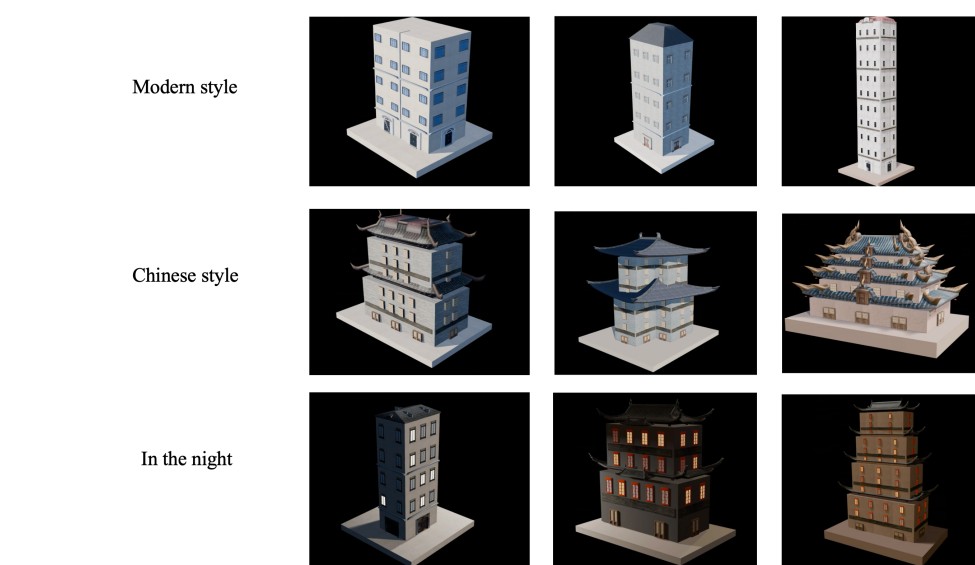

Modern style

Chinese style

In the night

Figure 7: **Generated Buildings Results.**

Each sample was assigned an $S_{\text{visual}}$ score through the aforementioned evaluation process. Training pairs were constructed by selecting sample pairs whose reward difference exceeded a predefined threshold, with the higher-scoring sample labeled as "chosen" and the lower-scoring as "rejected".

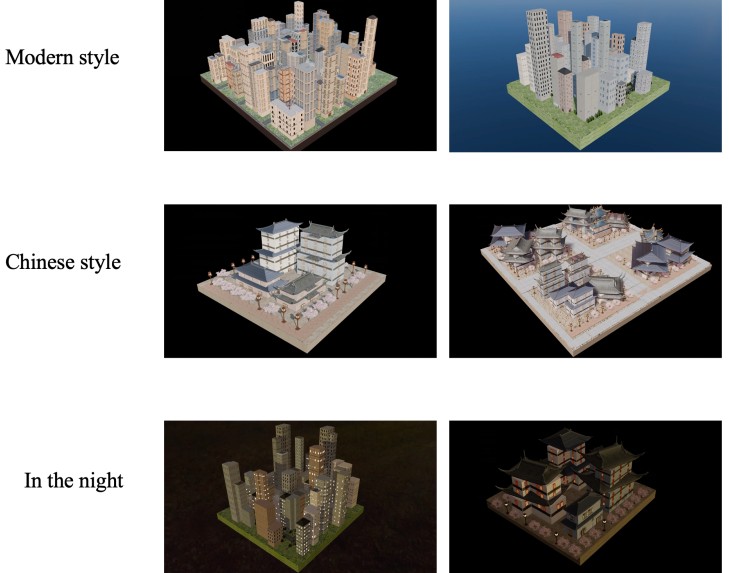

Modern style

Chinese style

In the night

Figure 8: **Generated City Results.**

# E TRAINING DETAILS

For BlockGen and BuidlingGen, we both adopt a two-stage training pipeline to fine-tune the Qwen3-8B model. Low-Rank Adaptation (LoRA) with a rank of 8 is applied to all target modules. The model is trained for 3 epochs with a batch size of 1, a gradient accumulation steps of 8, and a learning rate of $1 \times 10^{-4}$. A cosine learning rate scheduler with 10% warm-up is employed, and training is conducted in bfloat16 precision.

BlockGen's SFT was conducted on 4×NVIDIA A100 GPUs for approximately 5 hours, followed by RL where the reward model was trained for 10 minutes and the policy model optimized via PPO

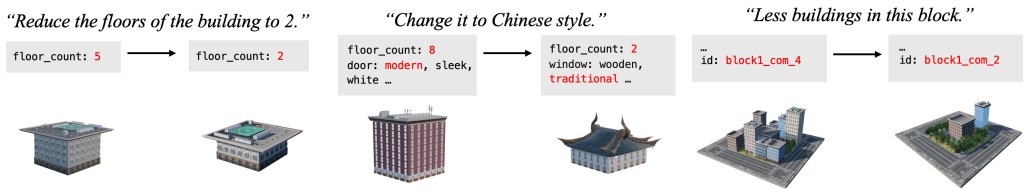

Figure 9: **Scene Manipulation Results.**

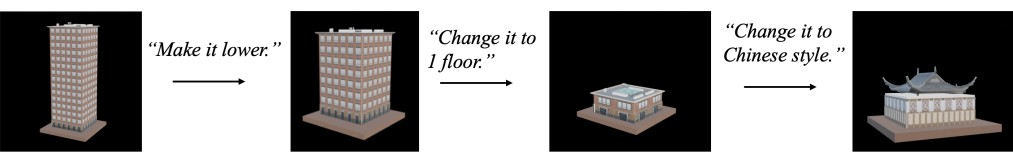

Figure 10: **Multi-step Manipulation Results.**

for 8 hours using the same LoRA and hyperparameter settings. BuildingGen's SFT was trained on 8×A100 GPUs for about 2 hours, and its RL stage involved 10 minutes of reward model training and 2 hours of PPO optimization, also under the same configuration.

## F   MORE GENERATED RESULTS

We present more results of buildings and scenes generated by CityGenAgent in Figure 7 and Figure 8.

## G   MANIPULATION

We provide additional examples of user interactions to demonstrate the manipulation ability of our model. As illustrated in Figure 9, these include precise control over the number of floor count, style switching, and block density adjustment. As shown in Figure 10, CityGenAgent can conduct multi-step manipulation based on the instruction. We present GPT and human evaluators with pairs of pre- and post-edit rendered images across 50 samples to assess instruction-following alignment and overall visual consistency. The resulting scores indicate that our editing approach effectively adheres to user instructions while maintaining global visual coherence, as shown in Table 5.

Table 5: **Quantitative Evaluation of Manipulation Performance.**

| Method | Text Alignment ↑ | | | Consisitency ↑ | | Success Rate ↑ |
|---|---|---|---|---|---|---|
| | CLIP | GPT | User | GPT | User | |
| CityGenAgent(Ours) | 0.286 | 8.7 | 8.4 | 6.6 | 5.9 | 98% |

## H   COMPARISON OF 3D CITY GENERATION.

As shown in Table 6, we summarize and compare 3D city scene generation methods in terms of text input, native 3D output, and supported functionalities. From the perspective of native 3D output, our approach directly produces 3D meshes, eliminating the need for post-processing steps such as converting NeRF-based video or point clouds into meshes—a process that often degrades 3D quality. This enables our results to be readily integrated into downstream applications. Furthermore, current city generation methods exhibit limitations in controllable editing. Our work addresses this gap by exploring user-driven customization, allowing the creation of novel assets.

Table 6: **Summary and Comparison of 3D City Generation.**

| Type | Method | Text Input | Native 3D Output | Creating New assets | Manipulation |
|---|---|---|---|---|---|
| Rendering-based | InifiCity (Lin et al., 2023) | × | NeRF | × | × |
| | CityDreamer (Xie et al., 2024) | × | NeRF | × | × |
| Diffusion-based | CityGen (Deng et al., 2025) | × | NeRF | × | × |
| | WonderJourney (Yu et al., 2024) | ✓ | Point Cloud | × | × |
| Procedure-based | 3D-GPT (Sun et al., 2025a) | ✓ | Mesh | × | × |
| | CityCraft (Deng et al., 2024) | ✓ | Mesh | × | × |
| | UrbanWorld (Shang et al., 2024) | ✓ | Mesh | ✓ | × |
| | CityGenAgent(Ours) | ✓ | Mesh | ✓ | ✓ |

# I ABLATION STUDY

To validate the effectiveness of our designed reward, we conduct the ablation study.

**Spatial Alignment Reward.** We examine how the reward affects the spatial plausibility of generated layouts. As reported in Table 7 and visualized in Figure 11, our full method achieves the best performance across all metrics. This demonstrates that the reward mechanism is critical for refining spatial relationships and ensuring physical plausibility, complementing the foundational capabilities learned through SFT. The reward signal provides essential, fine-grained spatial constraints that are difficult to learn from supervised data alone, effectively bridging the gap between high-level semantic correctness and low-level spatial realism.

**Visual Consistency Reward.** As shown in Figure 11 and Table 8, our full method achieves the highest scores in both Consistency and Text Alignment, demonstrating its effectiveness in generating coherent and semantically faithful building layouts. The notable gains occur after integrating the reward signal, which substantially improves *Text Alignment* by 0.7 points—a clear indication that the reward mechanism helps bridge the gap between the intermediate text-program representation and the final visual output. By optimizing with Visual Consistency Reward, the model is guided to transcend mere replication of the SFT data distribution, ultimately producing outputs that are not only semantically precise but also visually realistic.

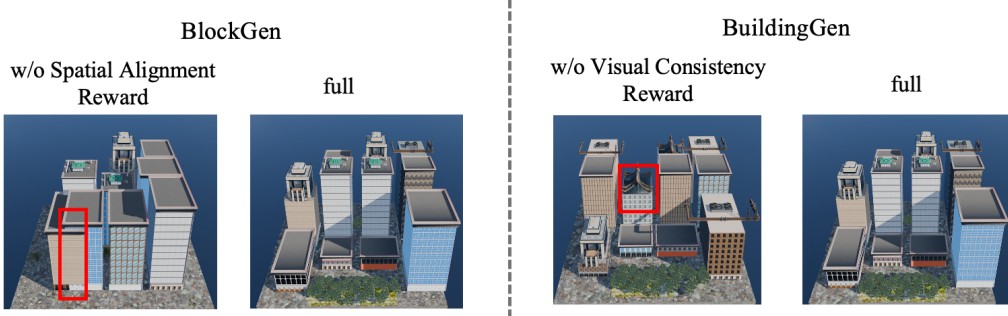

*User Instruction: The modern urban block consists of residential buildings, supermarkets, and 2 small green spaces. The residential buildings are evenly distributed throughout the block.*

Figure 11: **Ablation Study Results.** The red boxes highlight the model collision or style mismatch will otherwise occur without reward.

Table 7: **Ablation Study 1**. We evaluate the effect of Spatial Alignment Reward for BlockGen.

| Method | Collision ↓ | Pos. ↑ | PSA ↑ |
|---|---|---|---|
| w/o Spatial Alignment Reward | 5.59 | 80.17 | 84.02 |
| full | **4.89** | **85.33** | **87.90** |

Table 8: **Ablation Study 2**. We evaluate the effect of Visual Consistency Reward for BuildingGen.

| Method | Text Alignment ↑ | Consisency ↑ |
|---|---|---|
| w/o Visual Consistency Reward | 6.8 | 8.7 |
| full | **7.5** | **8.9** |

## J  PERFORMANCE CURVE

As shown in Figure 12, the curve in both subplots confirm that our iterative agent approach progressively enhances performance across multiple metrics. This validates the effectiveness of iterative refinement and RL-based optimization in achieving robust and consistent generation.

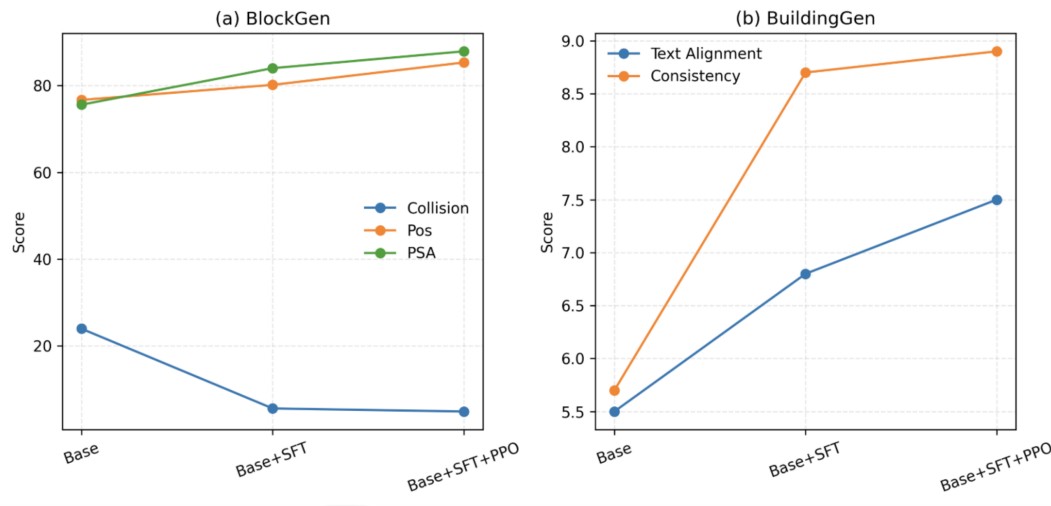

Figure 12: **Performance Curve.** We visualize the performance of BlockGen and BuildingGen during the finetuning process.

## K  LIMITATIONS

While CityGenAgent effectively generates high-quality 3D cities from natural language, several limitations remain: (i) Inference Time: The inference time can become relatively long for large-scale or highly complex scenes, as the system needs to process numerous blocks, buildings, and assets sequentially. (ii) Limited Generation Scale: The current system primarily focuses on generating city scenes at the block level, rather than modeling entire urban environments. This limitation primarily stems from the size and diversity of the synthetic dataset, which restricts the model's ability to generalize to larger and more complex city layouts.

## L  USAGE OF LLMS

LLMs were employed throughout the manuscript preparation exclusively for grammatical correction and stylistic refinements.

