# OpenReview forum: "Imagine a City: CityGenAgent for Procedural 3D City Generation"
_ICLR.cc/2026/Conference — Submitted to ICLR 2026_

### Official Review · Reviewer_xf2y · 2025-10-25

**Soundness:** 3
**Presentation:** 2
**Contribution:** 3
**Rating:** 4
**Confidence:** 4

**Summary:**

This paper presents CityGenAgent, a novel method for generating 3D cities through a natural language-based framework. The core contribution lies in developing domain-specific LLMs trained explicitly for city generation. The authors introduce two specialized programs—the Block Program for generating city blocks and the Building Program for describing individual buildings. They also construct a domain dataset for SFT and design multiple tailored rewards for RL training. Experimental results demonstrate that the proposed approach can produce high-quality 3D city layouts and exhibits strong controllability and manipulation capabilities.

**Strengths:**

1. Fine-tuning a pre-trained LLM for city generation is novel and interesting.
2. The design of the reward functions and the overall training objective is reasonable.
3. The proposed method achieves promising visual and structural results.

**Weaknesses:**

1. The abstract could be improved for clarity and conciseness. The current version is somewhat disorganized, making it difficult to grasp the core contribution. Based on the rest of the paper, the main novelty lies in developing domain-specific agents trained with SFT and RL, distinguishing this work from previous studies. The authors are encouraged to highlight this aspect more clearly in the abstract.
2. The process by which textual building descriptions are converted into 3D models is unclear. It would be helpful to specify whether the 3D assets are generated using a 3D generative model (e.g., Hunyuan3D) or retrieved from a pre-existing asset library.
3. The generation procedure for the training dataset is insufficiently described. Moreover, the authors do not commit to releasing the source code or dataset, and these materials are not provided as supplementary files. This lack of transparency makes it difficult to reproduce the proposed approach.
4. As a 3D generation study, the visual results are somewhat limited. Including video demonstrations would help validate the 3D spatial consistency and realism of the generated cities.
5. The authors could further explore alternative RL optimization strategies, such as DPO and GRPO, for further exploration.

**Questions:**

Will the authors release the source code and datasets?

---

> ### Author Response · Authors · 2025-11-21
> **Response to Reviewer xf2y (Part 1/2)**
>
> We sincerely thank Reviewer **xf2y** for the valuable comments provided during this review. The point-to-point responses are as follows.
>
> ---
>
> **W1:***"* The abstract could be improved for clarity and conciseness *. . ."*
>
> **A:** Thanks for your suggestion. We revised the abstract following your comments. The new version is as follows.
>
> The automated generation of interactive 3D cities is a critical challenge with broad applications in autonomous driving, virtual reality, and embodied intelligence. While recent advances in generative models and procedural techniques have improved the realism and scalability of city generation, existing methods often struggle with high-fidelity asset creation, controllability, and manipulation.
> In this work, we introduce CityGenAgent, a natural language-driven framework for hierarchical procedural generation of high-quality 3D cities. Our approach introduces two core programs, $\textbf{Block Program}$ and $\textbf{Building Program}$, which decompose city generation into interpretable and editable components.  To ensure structural correctness and semantic alignment, we adopt a two-stage learning strategy: (1) Supervised Fine-Tuning (SFT). We train $\textbf{BlockGen}$ and $\textbf{BuildingGen}$ to generate valid programs that adhere to schema constraints, including non-self-intersecting polygons and complete fields; (2) Reinforcement Learning (RL). We introduce Spatial Alignment Reward to improve spatial reasoning and Visual Consistency Reward to bridge the gap between textual descriptions and 3D realizations. Benefiting from program-based representation and models' generalization, CityGenAgent supports natural language editing and manipulation. Comprehensive evaluations demonstrate superior semantic alignment, visual quality, and controllability compared to existing methods, establishing a robust foundation for scalable 3D city generation.
>
> ---
>
> **W2:** *"*The process by which textual building descriptions are converted into 3D models is unclear*. . ."*
>
> **A:** Thank you for your comments. We construct an asset database including urban building elements collected from the Internet and segmented manually from complete building assets. The component descriptions generated by BuildingGen are matched with the most similar components in the database for instantiation. If the similarity falls below a predefined threshold (failure cases), we use Hunyuan3D-v2.5 to generate the target component.
>
> ---
>
> **W3:** *"*The generation procedure for the training dataset is insufficiently described. Moreover, the authors do not commit to releasing the source code or dataset, and these materials are not provided as supplementary files*. . ."*
>
> **A:** Thanks for your suggestion.
> The construction of training dataset for SFT and RL of two agents are presented in Appendix D.  We will release the codes and models upon publication.
>
> **The generation procedure for the training dataset:**
>
> -  For BlockGen, we curated 1,000 high-quality samples for supervised fine-tuning (SFT). Raw text–polygon pairs were post-processed to ensure validity. For RL preference data, we started with 1,000 block prompts, generated five candidates per prompt using GPT, scored them on a 0–10 scale, and selected the highest and lowest samples with a reward gap ≥5 to form 1,000 chosen–rejected pairs for PPO training.
>
> -  For BuildingGen, we constructed a paired dataset of 5,000 examples, each linking a natural language building description to its procedural program. To synthesize this, we collected 5,000 frontal building images from Google Maps and used GPT-4o to generate both holistic and component-level descriptions based on predefined architectural categories (e.g., door, window, roof). For reward model training, we gathered 5,000 diverse prompts, generating five samples per prompt to support preference-based learning.
>
> -  As for the asset library, we collected city and building components (such as windows, doors, roofs, and trees) collected from the Internet. In addition, we invited professional designers manually segmented building components from complete building models. Each asset undergoes manual inspection and normalization (e.g., orienting to face the +Y axis, moving to the origin) to ensure proper assembly in subsequent steps.
>
>
> ---
>
> **W4:** *"**As a 3D generation study, the visual results are somewhat limited**. . ."*
>
> **A:** Thank you for your suggestion. We add the video results in our webpage. [https://citygenagent.github.io/](https://citygenagent.github.io/) .

---

> > ### Author Response · Authors · 2025-11-21
> > **Response to Reviewer xf2y (Part 2/2)**
> >
> > **W5:** *"The authors could further explore alternative RL optimization strategies, such as DPO and GRPO, for further exploration."*
> >
> > **A:** Thank you for your suggestions. First, we explored DPO in our task. The metrics are as shown in the following tables. We can observe that applying PPO or DPO after SFT provides consistent improvements across all metrics, which further validates the effectiveness of our reward design.
> >
> > However, DPO performs slightly worse than PPO. The main reason is that our reward space is multi-dimensional, and PPO leverages a trained reward model that is more robust in handling these dimensions. In contrast, DPO learns preferences directly from positive and negative samples, which makes it harder to capture complex multi-dimensional signals.
> >
> > We also experimented with GRPO but were unable to achieve stable convergence. Unlike PPO, GRPO removes the value estimator and instead computes advantages using group-level statistics. While this design reduces memory and computational overhead, it is highly dependent on the reward variance within the group, which can lead to less stable training.
> >
> > BlokGen:
> >
> > | Method             | Collision ↓ | Pos. ↑  | PSA ↑   |
> > |--------------------|------------|---------|---------|
> > | Base Model         | 23.97%    | 76.70   | 75.60   |
> > | Base Model + SFT   | 5.59%     | 80.17   | 84.02   |
> > | Base Model + DPO   | 5.19%     | 81.13   | 85.03   |
> > | Base Model + PPO   | **4.89%** | **85.33** | **87.90** |
> >
> > BuildingGen:
> >
> > | Method             | Text Alignment ↑ | Consistency ↑ |
> > |--------------------|------------------|---------------|
> > | Base Model         | 5.5              | 5.7           |
> > | Base Model + SFT   | 6.8              | 8.7           |
> > | Base Model + DPO   | 7.0              | 8.1           |
> > | Base Model + PPO   | **7.5**              | **8.9**          |
> >
> > **Q:** *"Will the authors release the source code and datasets?"*
> >
> > **A:** We commit to release our source code upon publication.

---

> > ### Author Response · Authors · 2025-11-27
> > **Looking forward to hearing from you**
> >
> > Dear Reviewer xf2y：
> >
> > We sincerely appreciate your time and effort in reviewing our manuscript and providing valuable feedback.
> >
> > ---
> > In response to your insightful comments, we have made the following revisions to our manuscript and project page:
> >
> > * We have revised the abstract to highlight the core contribution.
> > * We have provided a more detailed breakdown of program execution and asset assembly in Section 3.4.
> > * We have added format accuracy to evaluate compliance of generated programs in Table 2.
> > * We have included a comparison with other language models (GPT-4o, Qwen2.5) in Table 2.
> > * We have added efficiency evaluation for per-scene generation in Section 4.2 and Table 3.
> > * We have published demo video results on our project webpage [https://citygenagent.github.io/](https://citygenagent.github.io/).
> > * We have added the multi-step results for manipulation in Figure 10.
> > * We have added the ablation study of RL methods (DPO and PPO) in Section 4.4.
> >
> > We hope these revisions adequately address your concerns.
> >
> > ---
> >
> > As the discussion phase is approaching its end,  we would be happy to address any additional points you may have during the remaining time of the discussion phase.
> >
> > ---
> >
> > Once again, we sincerely thank you for your contributions to improving the quality of this work.
> >
> > Best regards,
> >
> > The Authors of Submission 8493

---

### Official Review · Reviewer_z19P · 2025-10-28

**Soundness:** 2
**Presentation:** 3
**Contribution:** 2
**Rating:** 4
**Confidence:** 4

**Summary:**

This paper introduces CityGenAgent, a framework that integrates LLMs with hierarchical procedural generation techniques to create high-quality, interactive 3D city environments from natural language prompts. The core approach decomposes the complex city generation task into a structured hierarchy of specialized agents. The system seeks to improve controllability, fidelity, and manipulation capabilities of city generation.

**Strengths:**

The paper proposes to integrate LLMs to guide hierarchical procedural generation in the 3D domain, effectively addressing the traditional challenge of translating abstract intent into concrete geometric parameters. The results show the model can produce better 3D city scenes compared with existing methods.

**Weaknesses:**

1. The model efficiency is a critical weakness for a framework targeting real-world applications like large-scale city generation, which demand high efficiency. Given the reliance on commercial LLM APIs, the token consumption for a complete, complex city generation is likely prohibitive. The paper fails to provide a quantitative scaling analysis (e.g., generation time vs. area/asset count) or propose concrete technical solutions (beyond simple statements) to mitigate the LLM-related computational overhead.

2. The experimental validation focuses on the effect of certain rewards but critically omits the technical justification for the multi-agent hierarchy itself. Without an ablation study quantifying the marginal performance and token-efficiency gain of each agent division, the multi-agent design choice remains an arbitrary architectural decision rather than a proven necessity.

**Questions:**

1. What's the model used to create the asset?

2. Given the acknowledged limitation of inference time for large scenes, please provide quantitative scaling data: Generation time (in seconds or minutes) vs. scene complexity (measured by total area or asset count) for both CityGenAgent and baselines.

3. Will the performance of CityGenAgent be dependent on the reasoning and instruction-following capability of the underlying LLM? Have the authors experimented with different models?

4. It's encouraged to include some demos or videos to illustrate the generated 3D city scenes.

---

> ### Author Response · Authors · 2025-11-21
> **Response to Reviewer z19P**
>
> We sincerely thank  Reviewer **z19P** for recognizing the effectiveness of our approach in producing better 3D city scenes.  We now address your questions and concerns in detail.
>
> ---
>
> **W1:** *"*The paper fails to provide a quantitative scaling analysis*..."*
>
> **A:** Thanks for your comments. We have added the efficiency evaluation experiments, compared with Hunyuan3D, CityCraft and human craft . From the following table, we can observe that CityGenAgent is faster than other methods in generating the scene from a description for one block (100 x 100 m), maintaining a moderate memory usage. This efficiency demonstrates the potential of CityGenAgent to make 3D content creation significantly more streamlined and accessible.
>
> | Method        | Inference Time | VRAM  |
> |--------------|---------------|-------|
> | human        | 60 min        | -     |
> | Hunyuan3D    | 3 min         | 16GB  |
> | CityCraft    | 1 min         | -     |
> | CityGenAgent | **0.75 min**  | **8GB** |
>
> ---
>
> **W2:** *"...*an ablation study quantifying the marginal performance and token-efficiency gain of each agent division*. . ."*
>
> **A:** Thanks for spotting this. We added the token-efficiency evaluation as follows. CityGenAgent demonstrates a clear performance improvement compared to the single-agent Qwen3-8B, indicating that the our multi-agent design can achieve better task performance **while maintaining comparable token usage, thus improving overall token efficiency**.
>
> | Model         | Tokens | Performance | Token Efficiency |
> |--------------|--------|-------------|------------------|
> | Qwen3-8B      | 1089   | 77.83       | 7.15          |
> | CityGenAgent | **1134**   | **91.59**       | **8.08**             |
>
> ---
>
> **Q1:** *"What's the model used to create the asset?"*
>
> **A:** Thank you for your question.
> We construct an asset database including urban building elements collected from the Internet and segmented manually from complete building assets. The dataset contains city and building components (such as windows, doors, roofs, and trees) collected from the Internet. In addition, we invited professional designers to manually segment components from complete building models.
> Each asset undergoes manual inspection and normalization (e.g., orienting to face the +Y axis, moving to the origin) to ensure proper assembly in subsequent steps.
> The component descriptions generated by BuildingGen are matched with the most similar components in the database for instantiation. If the similarity falls below a predefined threshold (failure cases), we use Hunyuan3D-v2.5 to generate the target component.
>
> ---
>
> **Q2:** *"*please provide quantitative scaling data: Generation time (in seconds or minutes) vs. scene complexity*. . ."*
>
> **A:** Thanks for your comments. We compared the generation time of our approach under different scales, and the results show that CityGenAgent is capable of producing multi-block, large-scale scenes within a relatively short time.
>
> | Scale            | 100×100 m | 400×400 m |
> |------------------|-----------|-----------|
> | Generation time | 0.75 min  | 3 min     |
>
> ---
>
> **Q3:** *"*Will the performance of CityGenAgent be dependent on the reasoning and instruction-following capability of the underlying LLM? Have the authors experimented with different models?*"*
>
> **A:** Thank you for your question. The performance of CityGenAgent depends much more on *expert fine-tuning* than on the raw capabilities of the underlying LLM. We evaluated three strong base models (GPT-4o, Qwen2.5-7B, and Qwen3-8B), and none of them achieved satisfactory layout or building generation performance out of the box, even though their basic instruction-following abilities are similar after we unify the API through our DSL. The key performance gains come from fine-tuning the model into *specialized expert agents* for layout and building generation, which significantly improves spatial reasoning, constraint satisfaction, and collision avoidance. This indicates that reasoning ability is essential, but expert specialization (not the choice of base model) is the dominant factor behind CityGenAgent’s results.
>
> | Method                     | Format Accuracy | Collision | Pos.   | PSA    |
> |---------------------------|-----------------|-----------|--------|--------|
> | GPT-4o | 70%             | 6.67%     | 78.45  | 85.10  |
> | Qwen2.5-7B  | 70%             | 67.81%    | 67.60  | 61.25  |
> | Qwen3-8B  | 83%             | 23.97%    | 76.70  | 75.60  |
> | CityGenAgent w/o RL       | 98%             | 5.59%    | 80.17  | 84.02  |
> | CityGenAgent       | **98%**             | **4.89%**     | **85.33**  | **87.90**  |
>
> ---
>
> **Q4:** *"It's encouraged to include some demos or videos to illustrate the generated 3D city scenes."*
>
> **A:** Thank you for your suggestion. We added video results on our homepage [https://citygenagent.github.io/](https://citygenagent.github.io/) .

---

> ### Author Response · Authors · 2025-11-27
> **Looking forward to hearing from you**
>
> Dear Reviewer z19P：
>
> We sincerely appreciate your time and effort in reviewing our manuscript and providing valuable feedback.
>
> ---
> In response to your insightful comments, we have made the following revisions to our manuscript and project page:
>
> * We have revised the abstract to highlight the core contribution.
> * We have provided a more detailed breakdown of program execution and asset assembly in Section 3.4.
> * We have added format accuracy to evaluate compliance of generated programs in Table 2.
> * We have included a comparison with other language models (GPT-4o, Qwen2.5) in Table 2.
> * We have added efficiency evaluation for per-scene generation in Section 4.2 and Table 3.
> * We have published demo video results on our project webpage [https://citygenagent.github.io/](https://citygenagent.github.io/).
> * We have added the multi-step results for manipulation in Figure 10.
> * We have added the ablation study of RL methods (DPO and PPO) in Section 4.4.
>
> We hope these revisions adequately address your concerns.
>
> ---
>
> As the discussion phase is approaching its end,  we would be happy to address any additional points you may have during the remaining time of the discussion phase.
>
> ---
>
> Once again, we sincerely thank you for your contributions to improving the quality of this work.
>
> Best regards,
>
> The Authors of Submission 8493

---

### Official Review · Reviewer_v4kQ · 2025-10-30

**Soundness:** 3
**Presentation:** 3
**Contribution:** 3
**Rating:** 6
**Confidence:** 4

**Summary:**

This work presents a natural language-driven framework named CityGenAgent for hierarchical generation of high-quality 3D city models. The framework is based on large language models and decomposes the city generation process into two core programs, Block Program and Building Program, which are executed by the BlockGen and BuildingGen modules respectively. The paper designs spatial alignment reward and visual consistency reward to enhance spatial reasoning and visual fidelity, and supports interactive operations through natural language.

**Strengths:**

1. The paper introduces a novel hierarchical procedural generation paradigm via two domain-specific languages (Block Program and Building Program), creatively combining LLMs with structured programs as editable proxies.

2. The technical execution is robust, with clear decomposition into BlockGen/BuildingGen modules, SFT+PPO training pipeline, and reward designs grounded in computable metrics, demonstrating better semantic alignment and visual fidelity over mentioned baselines.

3. The paper is well-structured with intuitive figures, precise DSL definitions, and appendices for prompts and datasets.

4. This advances embodied AI and simulation by providing a scalable, controllable foundation for 3D urban worlds, potentially inspiring similar program-based agents in other procedural domains

**Weaknesses:**

1. The framework in Section 3.1 decomposes cities using Block Program and Building Program as editable DSL intermediates. However, no comparison is provided with scene graph-based 3D scene generation methods, such as in terms of layout fidelity, editing efficiency, or scalability to multi-block cities.

2. In Section 3.2.1, Block-Gen (SFT) is described as enabling the LLM to generate valid Block Programs that adhere to the schema, including non-self-intersecting polygons and required fields. However, no compliance metrics, such as program parse success rate after natural language conversion, field completeness ratio, or polygon validity, are reported, leaving the structural integrity of the DSL outputs unquantified prior to 3D execution.

3. In Section 3.2.2, Block-Gen(PPO) is introduced to improve spatial reasoning over SFT via Spatial Alignment Reward, claiming better handling of complex layouts. Yet, no ablation study compares SFT-only versus SFT+PPO on PSA or Collision Rate, making it unclear how large the performance gap is and whether PPO is necessary for core gains or merely a non-essential optimization.

4. The reliance on synthetic data generated by GPT-4o for SFT/PPO (Appendix D) limits real-world generalization; while post-processing removes low-quality samples, the dataset may not capture diverse urban styles (e.g., non-Western architectures) or corner cases like irregular terrains or imbalanced building attributes.

**Questions:**

1. For Spatial Alignment Reward, why choose AABBs over exact polygon overlap? Share ablation results on reward sensitivity to this approximation, and what if switching to exact methods.

2. In program execution (Section 3.4), how are assets for Building Program components sourced or synthesized when no match exists? Provide failure cases handling and success rates from your 50 evaluation prompts.

3. The intro mentions embodied intelligence applications—have you tested any integration with simulators? what adaptations are needed?

4. Manipulation (Section 3.5) claims RL enables generalization—please quantify this with metrics like edit success rate, coherence preservation, and multi-step edit chains.

---

> ### Author Response · Authors · 2025-11-21
> **Response to Reviewer v4kQ (Part1/3)**
>
> We are grateful to Reviewer v4kQ for the thoughtful assessment and positive evaluation of our work. We now address your questions and concerns in detail.
>
> ---
>
> **W1:** *"...*no comparison is provided with scene graph-based 3D scene generation methods*..."*
>
> **A:** Thanks for your comments. We focus on outdoor scene-graph methods, which are more relevant to our task. Two representative works (*Controllable 3D Outdoor Scene Generation via Scene Graphs [1] (ICCV 2025) ​* and *SceneCraft* [2] (ICML 2024)) both reveal clear limitations: they rely on coarse voxelized or template-based geometry, produce low-fidelity outdoor structures (e.g., buildings, roads, façades), and do not scale beyond small scenes due to graph size explosion and expensive global rewiring. Besides, both of them do not release codes, making direct quantitative comparison infeasible
>
> In contrast, our Block and Building Programs introduce a hierarchical, procedural DSL that naturally captures urban regularities (road loops, parcel grids, facade repetition) and enables local, efficient editing. This decomposition avoids the combinatorial complexity of large scene graphs and allows scalable generation of multi-block cities, which existing scene-graph methods have not demonstrated.
>
> We will include citations to these two works [1][2] in our paper to clarify the context and highlight the differences.
>
> ---
>
> **References**
>
> [1] Liu, Yuheng, et al. "Controllable 3D outdoor scene generation via scene graphs." *arXiv preprint arXiv:2503.07152* (2025).
>
> [2] Hu, Ziniu, et al. "Scenecraft: An llm agent for synthesizing 3d scenes as blender code." ​*Forty-first International Conference on Machine Learning*​. 2024.
>
> ---
>
> **W2:** *"*no compliance metrics, such as program parse success rate after natural language conversion*. . ."*
>
> **A:** Thank you for spotting this.
> To quantitatively assess whether the generated program complies with the specification, we adopt a metric termed Format Accuracy. This evaluation encompasses three key criteria:
> (i) the validity of the output as a parsable JSON structure
> (ii) the geometric correctness of polygon definitions
> (iii) the completeness of required fields within the generated programs.
>
> From the quantitative results, the closed-source model (GPT-4o) and open-source models such as Qwen2.5 and Qwen3 exhibit format accuracy below 90%. After SFT stage, the output format accuracy improves to 98%, providing a strong guarantee for reliable execution of subsequent program components.
>
> | Method                     | Format Accuracy | Collision | Pos.   | PSA    |
> |---------------------------|-----------------|-----------|--------|--------|
> | GPT-4o  | 70%             | 6.67%     | 78.45  | 85.10  |
> | Qwen2.5-7B  | 70%             | 67.81%    | 67.60  | 61.25  |
> | Qwen3-8B   | 83%             | 23.97%    | 76.70  | 75.60  |
> | CityGenAgent w/o RL       | 98%             | 5.59%    | 80.17  | 84.02  |
> | CityGenAgent       | **98%**             | **4.89%**     | **85.33**  | **87.90**  |
>
> To deal with the 2% error and ensure the fairness of comparison, we retry until a properly formatted program is produced that can be parsed by the executor to extract valid parameters.
>
> ---
>
> **W3:** *"...* no ablation study compares SFT-only versus SFT+PPO on PSA or Collision Rate *..."*
>
> **A:** Thanks for your insightful suggestion!
> We conducted the corresponding ablation studies, which are presented in Section 4.4. We present the table here. After RL, our model achieves the best performance across all metrics. This demonstrates that the reward mechanism is critical for refining spatial relationships and ensuring physical plausibility, complementing the foundational capabilities learned through SFT. The reward signal provides essential, fine-grained spatial constraints that are difficult to learn from supervised data alone, effectively bridging the gap between high-level semantic correctness and low-level spatial realism.
>
> | Method             | Collision ↓ | Pos. ↑  | PSA ↑   |
> |--------------------|------------|---------|---------|
> | Base Model         | 23.97%    | 76.70   | 75.60   |
> | Base Model + SFT   | 5.59%     | 80.17   | 84.02   |
> | Base Model + DPO   | 5.19%     | 81.13   | 85.03   |
> | Base Model + PPO   | **4.89%** | **85.33** | **87.90** |
>
> ---

---

> ### Author Response · Authors · 2025-11-21
> **Response to Reviewer v4kQ  (Part2/3)**
>
> **W4:** *"...* the dataset may not capture diverse urban styles (e.g., non-Western architectures) or corner cases like irregular terrains or imbalanced building attributes *..."*
>
> **A:** Thank you for the reviewer’s constructive comments. We agree that synthetic datasets may risk insufficient coverage of certain architectural styles or long-tail spatial configurations. We clarify that CityGenAgent is not constrained by the distribution of the synthetic geometry data, for the following reasons.
>
> - **Diversity of layout data beyond simple shapes**
>
> Although the dataset is synthetic, layout diversity is achieved through LLM-driven stochastic generation rather than handcrafted operations. Our prompts incorporate varied zoning schemes, irregular parcel boundaries, mixed-use patterns, and heterogeneous density profiles, enabling GPT to generate highly diverse block structures. In addition, we adopt an automatic filtering system based on geometric overlap, footprint density, and global plausibility to retain diverse yet physically realistic samples. Importantly, the Spatial Alignment Reward introduces geometry-level inductive biases—such as non-overlap, reasonable spacing, and structural feasibility—that are style-independent and universally valid across global urban contexts. Since PPO optimizes for these universal constraints rather than the empirical distribution of the synthetic dataset, BlockGen is able to generalize to spatial arrangements significantly beyond the training distribution.
>
> - **Architectural style diversity.**
>
> From the input description of the scene style, CityGenAgent can generate different architectural styles, as shown in Figure 4 in the paper. We can produce modern and traditional Chinese style scenes. Extending to more styles can be easily achieved by adding more assets for asset retrieval, which requires no any additional training.
>
> - **RL improves the model’s generalization ability**
>
> Although synthetic datasets have inherent limitations, we enhance the model’s generalization ability during the RL stage by incorporating execution-based feedback into the reward function. The reward signals derived from actual execution outcomes not only prevent the model from relying on rote memorization of the training set, but also encourage it to learn more robust and transferable behaviors across a broader state space, as noted in [1]. This improved generalization is further reflected in the model’s manipulation capability: even when the model has not encountered certain cases in the training data, it can still perform the required edits successfully according to the instructions, demonstrating strong adaptability to unseen scenarios.
>
> ---
> **References**
>
> [1] Chu, Tianzhe, et al. "Sft memorizes, rl generalizes: A comparative study of foundation model post-training." arXiv preprint arXiv:2501.17161 (2025).
>
> ---
>
> **Q1:** *"...*Share ablation results on reward sensitivity to this approximation, and what if switching to exact methods.*"*
>
> **A:** We appreciate your suggestion to consider exact polygon overlap.
> The experimental results demonstrate the advantages of using AABB overlap over exact polygon intersection during training.. This indicates that AABB provides a more conservative and effective collision penalty, improving spatial safety.
> We present it from three technical considerations:
>
> - **Numerical stability under vertex jitter**
>   During training, polygon vertices can      exhibit small perturbations due to iterative refinement. Exact polygon      intersection is highly sensitive to such jitter, often producing unstable      or discontinuous penalty signals.
> - **Conservative upper bound**
>   AABB overlap provides an upper bound on true polygon intersection. We prefer to slightly over-penalize potential collisions rather than under-penalize them. This conservative approach improves safety and spatial feasibility during generation.
> - **Computation**
>   Additionally, AABB computation is more efficient and scales better with scene complexity, whereas polygon intersection requires expensive geometric operations and introduces substantial runtime overhead.
>
> | Method         | Collision | Pos.    | PSA    |
> |---------------|-----------|---------|--------|
> |Exact Polygon|5.57%|80.70|82.62|
> | AABBs    | **4.89%** | **85.33** | **87.90** |
>
> ---
>
> **Q2:** *"*how are assets for Building Program components sourced or synthesized when no match exists? Provide failure cases handling and success rates from your 50 evaluation prompts.*"*
>
> **A:** Thanks for your question. We retrieves the most similar components from the resource library based on text similarity for assembly. If the similarity falls below a predefined threshold (failure cases), we use Hunyuan3D-v2.5 to generate the target component.

---

> ### Author Response · Authors · 2025-11-21
> **Response to Reviewer v4kQ (Part3/3)**
>
> **Q3:** *"*The intro mentions embodied intelligence applications—have you tested any integration with simulators? what adaptations are needed?*"*
>
> **A:** We can integrate the scenes we generate as scene assets into Unreal Engine (UE) to serve as simulation environments. You can check it in our homepage  https://citygenagent.github.io/  . Since the scenes we generate are of high quality and highly realistic, with well-structured geometry, they can be seamlessly applied to tasks such as simulation environment development and related applications.
>
> **Q4:** *"*Manipulation (Section 3.5) claims RL enables generalization—please quantify this with metrics like edit success rate, coherence preservation, and multi-step edit chains.*"*
>
> **A:** Thank you for your question. The quantatitive results for editing the scene are shown here. We also added the multi-step results in Figure 10 in the paper.
>
> | Method        | CLIP   | GPT   | User  | Successful Rate |
> |--------------|--------|-------|-------|-----------------|
> | CityGenAgent | 0.286  | 8.7   | 8.4   | 98%             |

---

> ### Author Response · Authors · 2025-11-27
> **Looking forward to hearing from you**
>
> Dear Reviewer v4kQ：
>
> We sincerely appreciate your time and effort in reviewing our manuscript and providing valuable feedback.
>
> ---
> In response to your insightful comments, we have made the following revisions to our manuscript and project page:
>
> * We have revised the abstract to highlight the core contribution.
> * We have provided a more detailed breakdown of program execution and asset assembly in Section 3.4.
> * We have added format accuracy to evaluate compliance of generated programs in Table 2.
> * We have included a comparison with other language models (GPT-4o, Qwen2.5) in Table 2.
> * We have added efficiency evaluation for per-scene generation in Section 4.2 and Table 3.
> * We have published demo video results on our project webpage [https://citygenagent.github.io/](https://citygenagent.github.io/).
> * We have added the multi-step results for manipulation in Figure 10.
> * We have added the ablation study of RL methods (DPO and PPO) in Section 4.4.
>
> We hope these revisions adequately address your concerns.
>
> ---
>
> As the discussion phase is approaching its end,  we would be happy to address any additional points you may have during the remaining time of the discussion phase.
>
> ---
>
> Once again, we sincerely thank you for your contributions to improving the quality of this work.
>
> Best regards,
>
> The Authors of Submission 8493

---

### Official Review · Reviewer_LEo5 · 2025-11-02

**Soundness:** 3
**Presentation:** 2
**Contribution:** 3
**Rating:** 4
**Confidence:** 4

**Summary:**

This paper, Imagine a City: CityGenAgent for Procedural 3D City Generation, proposes a natural language-driven framework, CityGenAgent, which leverages large language models (LLMs) for the hierarchical procedural generation of 3D cities. The core contribution is using an agent-based system to iteratively refine and generate complex 3D scenes based on descriptive text input, aiming to improve controllability and fidelity in automated 3D content creation.

**Strengths:**

1. The task of procedural, language-guided 3D city generation is interesting, addresses a crucial challenge in 3D content creation, and is of interest to the community.

2. The paper is generally well-organized and the approach is presented clearly, making the high-level idea easy for the reader to follow and understand.

**Weaknesses:**

1. The technical contribution appears somewhat limited. While the paper focuses heavily on the scene description and iterative refinement driven by the LLM, there is a distinct lack of detailed description regarding the actual procedural generation and manipulation of the 3D assets (e.g., buildings, road networks, and underlying geometric operations). For a 3D generation work, the mechanisms for handling 3D assets should be a core component, yet these sections lack sufficient technical detail.

2. The experimental validation is not enough to scientifically support the claims. The majority of the results presented are qualitative demonstrations, which, while visually appealing, are difficult to compare scientifically and objectively. The most crucial quantitative results are confined primarily to a single table (presumably Table 1). The quantitative evaluation is conducted on a very small sample size (only 50 samples), which significantly undermines the statistical persuasiveness and generalizability of the reported results. The comparison against existing methods appears sparse, making it difficult to properly contextualize the performance and novelty of the proposed CityGenAgent.

3. The paper does not include any discussion or data regarding the generation efficiency, computational cost, or inference time, which are critical factors for a procedural generation system, especially one leveraging large models.

**Questions:**

1. Please provide a more complete and detailed breakdown of the 3D generation process. Specifically, elaborate on how the refined scene descriptions are translated into concrete 3D asset instantiation, placement, and manipulation operations. This is essential for readers to understand and potentially replicate the method.

2. The experimental section requires substantial expansion. This must include: Expanding the size of the quantitative evaluation dataset significantly to ensure statistical validity;  Including more competitive and relevant baseline methods for comprehensive performance comparison; Providing quantitative analysis on the computational costs, such as inference time and memory usage, particularly in relation to the size and complexity of the generated scene.

3. To better illustrate the value of the proposed iterative agent approach, please provide a detailed experimental analysis showing the progressive improvement over the training or generation iterations (e.g., convergence curves or metric improvements over steps). This would help demonstrate why the iterative description refinement is necessary and effective.

---

> ### Author Response · Authors · 2025-11-21
> **Response to Reviewer LEo5 (Part 1/2)**
>
> We sincerely thank  Reviewer **LEo5** for recognizing the novelty and relevance of our work on procedural 3D city generation. We appreciate your constructive feedback and now address your concerns and questions in detail.
>
> ---
>
> **W1:** *"...*The technical contribution appears somewhat limited … the mechanisms for handling 3D assets should be a core component*..."*
>
> **A:** Thanks for your comments.
>
> - **The technical contribution appears somewhat limited.**
>
>   First, let’s clarify that our contributions. We introduce two core programs (Block Program and Building Program) which decompose city generation into interpretable and editable components. To ensure structural correctness and semantic alignment, we adopt a two-stage learning strategy: (1) Supervised Fine-Tuning (SFT). We train BlockGen and BuildingGen to generate valid programs that adhere to schema constraints, including non-self-intersecting polygons and complete fields; (2) Reinforcement Learning (RL). We introduce Spatial Alignment Reward to improve spatial reasoning and Visual Consistency Reward to bridge the gap between textual descriptions and 3D realizations. Benefiting from program-based representation and models' generalization, CityGenAgent supports natural language editing and manipulation.
>
>   While procedural generation workflows are well-established in video games and modeling, our research emphasizes creating efficient and semantically consistent parameterized representations that seamlessly integrate into these workflows to explore how to enhance efficiency in 3D content creation.
> - **The mechanisms for handling 3D assets.**
>
>   After obtaining the program, the 3D scene generation process consists of two stages: asset preparation and asset assembly.
>
>   - In the​**​ asset preparation stage**​, we retrieve or use Hunyuan3D to generate the asset . For retrieval, inspired by HOLODECK, we match the component descriptions generated by BuildingGen with the most similar components in the database for instantiation. If the similarity score falls below a threshold, we use Hunyuan3D to generate a new asset based on the component description. In our experiment, the asset database contains  city and building components (such as windows, doors, roofs, and trees) collected from the Internet. In addition, we invited professional designers manually segmented  building components from complete building models. Each asset undergoes manual inspection and normalization (e.g., orienting to face the +Y axis, moving to the origin) to ensure proper assembly in subsequent steps.
>   - In the ​**procedural assembly stage**​, we follow standard procedural generation principles. Parameters provided by BlockGen determine the position and orientation of each building edge, which in turn defines the transformations for components on that wall. Our executor leverages Blender’s API, encapsulating operations such as rotation, scaling, and placement to assemble assets.
>
> ---

---

> ### Author Response · Authors · 2025-11-21
> **Response to Reviewer LEo5 (Part 2/2)**
>
> **W2:** *"The quantitative evaluation is conducted on a very small sample size  . . .  The comparison against existing methods appears sparse . . ."*
>
> **A:** Thanks for your valuable feedback.
>
> - **A very small sample size.**
>
>   In designing our evaluation, we follow the experimental settings commonly adopted in procedural generation works. For instance, SceneCraft [1] evaluates 40 prompts for general scene generation, while CityX [2] and SceneX [3] use 50 prompts for scene generation. Given that 3D city scenes involve a much larger spatial extent and richer content, we test 50 diverse scenes to assess the proposed metrics. To ensure diversity, we designed two types of input prompts:
>
>   - Coarse-grained descriptions, like "The block contains 4 high-rise buildings with small gardens interspersed."
>   - Fine-grained descriptions , like "The urban block contains 14 mid-rise residential buildings occupying most of the area, 2 small commercial structures taking up 20% of the space, and green spaces comprising the remaining space. The layout features a large park in the center with buildings along the perimeter" .
>
>     All generated samples were further included in a user study, where human evaluators provided scores and feedback to guarantee that every test case was assessed effectively, as detailed in Appendix B.
>
> - **Comparison against existing methods appears sparse**
>
>   We have added comparisons with most recent three procedural scene generation methods: UrbanWorld, 3DGPT, and SceneCraft. UrbanWorld focuses on urban scene generation, while 3DGPT and SceneCraft target general scenes. We evaluate their results from a visual perspective using VLM-based scoring and a user study ( evaluation details in Appendix B), enabling a more comprehensive comparison with our city scene generation approach. Our method achieves higher scores in both GPT-based evaluation and user study.
>
> | Method        | GPT  | User Study |
> |--------------|------|------------|
> | UrbanWorld   | 5.5  | 4.5        |
> | 3D-GPT       | 6.6  | 5.5        |
> | SceneCraft   | 6.0  | 5.2        |
> | Hunyuan3D    | 6.5  | 5.5        |
> | CityCraft    | 6.1  | 5.1        |
> | CityGenAgent | **6.7** | **5.8** |
>
> ---
>
> **References**
>
> [1] Hu, Ziniu, et al. "Scenecraft: An llm agent for synthesizing 3d scenes as blender code." ​*Forty-first International Conference on Machine Learning*​. 2024.
>
> [2] Zhang, Shougao, et al. "Cityx: Controllable procedural content generation for unbounded 3d cities." *arXiv preprint arXiv:2407.17572* (2024).
>
> [3] Zhou, Mengqi, et al. "SceneX: Procedural Controllable Large-scale Scene Generation." *arXiv preprint arXiv:2403.15698* (2024).
>
> ---
>
> **W3:** *"...*  the generation efficiency, computational cost, or inference time, which are critical factors for a procedural generation system *..."*
>
> **A:** Thank you for noting this weakness; we have addressed it accordingly. We add the computation costs experiments, compared with Hunyuan3D, CityCraft and human craft . From the following table, we can observe that CityGenAgent is faster than other methods in generating the scene from a description for one block (100 x 100 m), maintaining a moderate memory usage. This efficiency demonstrates the potential of CityGenAgent to make 3D content creation significantly more streamlined and accessible.
>
> | Method        | Inference Time | VRAM  |
> |--------------|---------------|-------|
> | human        | 60 min        | -     |
> | Hunyuan3D    | 3 min         | 16GB  |
> | CityCraft    | 1 min         | -     |
> | CityGenAgent | **0.75 min**  | **8GB** |
>
> ---
>
> **Q1:** *"...* how the refined scene descriptions are translated into concrete 3D asset instantiation, placement, and manipulation operations *..."*
>
> **A:** Thank you for your question.
> After obtaining the program, the 3D scene generation process consists of two stages: asset preparation and asset assembly. The details are present in the response for Weakness 1.
>
> ---
>
> **Q2:** *"*The experimental section requires substantial expansion.*"*
>
> **A:**  Thank you for your valuable feedback. We address the concern in the response of Weakness 2 and 3.
> - We added more relevant methods (UrbanWorld, 3DGPT, SceneCraft) for performance comparison.
> - We provided quantitative analysis on the inference time and memory usage for one block (100m x 100m).
> ---
>
> **Q3:** *"* showing the progressive improvement over the training or generation iterations ...*"*
>
> **A:** Thank you for your asking. We visualized the performance curves of the base model and the fine-tuned model in Appendix J . The curve in both subplots confirm that our iterative agent approach progressively enhances performance across multiple metrics. This validates the effectiveness of iterative refinement and RL-based optimization in achieving robust and consistent generation.

---

> ### Author Response · Authors · 2025-11-27
> **Looking forward to hearing from you**
>
> Dear Reviewer LEo5：
>
> We sincerely appreciate your time and effort in reviewing our manuscript and providing valuable feedback.
>
> ---
> In response to your insightful comments, we have made the following revisions to our manuscript and project page:
>
> * We have revised the abstract to highlight the core contribution.
> * We have provided a more detailed breakdown of program execution and asset assembly in Section 3.4.
> * We have added format accuracy to evaluate compliance of generated programs in Table 2.
> * We have included a comparison with other language models (GPT-4o, Qwen2.5) in Table 2.
> * We have added efficiency evaluation for per-scene generation in Section 4.2 and Table 3.
> * We have published demo video results on our project webpage [https://citygenagent.github.io/](https://citygenagent.github.io/).
> * We have added the multi-step results for manipulation in Figure 10.
> * We have added the ablation study of RL methods (DPO and PPO) in Section 4.4.
>
> We hope these revisions adequately address your concerns.
>
> ---
>
> As the discussion phase is approaching its end,  we would be happy to address any additional points you may have during the remaining time of the discussion phase.
>
> ---
>
> Once again, we sincerely thank you for your contributions to improving the quality of this work.
>
> Best regards,
>
> The Authors of Submission 8493

---

### Author Response · Authors · 2025-11-21
**General Response**

We are encouraged to see that our reviewers recognize this work as a novel and high-quality framework for language-guided 3D city generation:

- Reviewers LEo5 and v4kQ think our paper is well-written, clearly organized, and easy to follow, with intuitive figures and precise DSL definitions.
- Reviewers z19P, v4kQ, and xf2y highlight that our approach is novel, integrating LLMs with hierarchical procedural generation and fine-tuning strategies to bridge abstract intent and concrete geometry.
- Reviewers v4kQ and xf2y appreciate the robust technical execution, including the BlockGen/BuildingGen decomposition, SFT+PPO pipeline, and reward design grounded in computable metrics.
- Reviewers z19P and xf2y agree that our method achieves promising visual and structural results, outperforming existing baselines in semantic alignment and visual fidelity.

---

We highlight our contributions as follows:

We introduce two core programs—**Block Program** and **Building Program**—which decompose city generation into interpretable and editable components. To ensure structural correctness and semantic alignment, we adopt a two-stage learning strategy: **SFT and RL** to improve spatial reasoning and visual consistency. Based on our program-based representation and the model’s generalization, **CityGenAgent** supports natural language editing and manipulation.

---

As suggested by the reviewers, we revised our manuscript as follows:

- Revised the abstract to highlight the core contribution, as suggested by Reviewer xf2y.
- Provided a more detailed breakdown of program execution and asset assembly in Section 3.4, as suggested by Reviewer LEo5, xf2y and z19p.
- Added format accuracy to evaluate compliance of generated programs in Table 2, as suggested by Reviewer v4kQ.
- Included a comparison with other language models in Table 2, as suggested by Reviewer z19P.
- Added efficiency evaluation for per-scene generation in Section 4.2 and Table 3, as suggested by Reviewers z19P and v4kQ.
- Published demo video results on our project webpage, as suggested by Reviewers z19P and v4kQ.
- Added the multi-step results in Figure 10, as suggested by Reviewers v4kq and xf2y.
- Added the ablation study of RL methods  in Section 4.4 ,as suggested by Reviewers xf2y and v4kQ.

---

### Author Response · Authors · 2025-11-29
**A Summary Comment to the New Area Chair**

Dear Area Chair,

We sincerely thank the Area Chair for reviewing our paper and providing valuable oversight.

---

We are encouraged that the reviewers recognize our work as a well-written (Reviewers LEo5 and v4kQ ), novel (Reviewers LEo5, z19P, v4kQ, and xf2y) and high-quality framework (Reviewers z19P and xf2y) for language-guided 3D city generation.

The reviewers’ concerns primarily focus on supplementing experiments and providing detailed explanations, without questioning the novelty or performance of our approach. We first summarize  our reviewers' common concerns and our responses, and then present a brief response to reviewers' other specific questions.

---
**Common Concerns**
- **Visual Results** (Suggested by Reviewers xf2y, z19P and v4kQ)

  we have published demo video results on our project webpage [https://citygenagent.github.io/](https://citygenagent.github.io/) to help validate the 3D spatial consistency and realism of the generated cities.
- **Efficiency evaluation for scene generation** (Suggested by Reviewers z19P and v4kQ)

  We have added a comparative evaluation of inference time and GPU memory consumption across different methods and the generation time of our method for a single block (100×100m) and multiple blocks (400×400m) .

  | Method        | Generation Time | VRAM  |
  |--------------|---------------|-------|
  | human        | 60 min        | -     |
  | Hunyuan3D    | 3 min         | 16GB  |
  | CityCraft    | 1 min         | -     |
  | CityGenAgent | **0.75 min**  | **8GB** |

  | Scale            | 100×100 m | 400×400 m |
  |------------------|-----------|-----------|
  | Generation time | 0.75 min  | 3 min     |
- **3D Asset Sourcing** (Suggested by Reviewers LEo5, xf2y and z19p)

  We manually construct an asset database containing urban building elements. The 3D component assets are instantiated either by retrieving them from the database or by generating them with Hunyuan3D-v2.5 when their similarity falls below a predefined threshold.
---

**Response to Reviewer LEo5’s Comments**

Reviewer LEo5 concerns the technical detail of procedural generation of the 3D assets, expansion of test set size.

First, let's clarify our technical contribution. While procedural generation workflows are well-established in video games, our research emphasizes **injecting domain knowlege to LLMs to generate efficient and semantically consistent parameterized representations (Block Program and Building Program)** that seamlessly integrate into these workflows.

Moreover, given that 3D city scenes involve richer content, we follow the experimental  common settings adopted in procedural generation works (SceneCraft, CityX, SceneX), which test around 50 scenes to assess the proposed metrics.

---

**Response to Reviewer v4kQ’s Comments**

Reviewer v4kQ concerns the compliance metrics and the reliance on synthetic data. In response，

- We have added the evaluation results on Format Accuracy to quantitatively assess whether the generated program complies with the specification.

  | Method                     | Format Accuracy | Collision | Pos.   | PSA    |
  |---------------------------|-----------------|-----------|--------|--------|
  | GPT-4o  | 70%             | 6.67%     | 78.45  | 85.10  |
  | Qwen2.5-7B  | 70%             | 67.81%    | 67.60  | 61.25  |
  | Qwen3-8B   | 83%             | 23.97%    | 76.70  | 75.60  |
  | CityGenAgent w/o RL       | 98%             | 5.59%    | 80.17  | 84.02  |
  | CityGenAgent       | **98%**             | **4.89%**     | **85.33**  | **87.90**  |

- We have explained how we address the limitations of synthetic data by considering the key factors: dataset construction and the generalization capability introduced by reinforcement learning.

---
**Summary for the Area Chair**

Importantly, none of the issues raised by our reviewers challenge the soundness or novelty of our approach. All concerns have been fully addressed through:

- provided visual demo results,
- additional experiments (including inference time, compliance metrics and ablation study),
- clearer explanations of our method,
- revised sections to improve clarity.

We highlight our contributions as follows:

We introduce two core programs—**Block Program** and **Building Program**—which decompose city generation into interpretable and editable components. To generate these programs, we adopt a two-stage training strategy: SFT and RL. In the SFT stage, the model learns to ensure format correctness and semantic alignment, while RL further enhances spatial reasoning and visual consistency. Based on our program-based representation and the model’s generalization, our framework supports natural language editing and manipulation. Comprehensive evaluations show that CityGenAgent achieves impressive semantic alignment and higher visual quality, establishing a stronger foundation for broad applications.

---
We sincerely appreciate the reviewers’ effort and the AC’s consideration.

---

### Meta-Review · Area_Chair_85Pa · 2025-12-27

**Summary:**

Reviewers generally find the paper technically solid and well executed, presenting a novel framework for language-guided procedural 3D city generation with strong controllability and visually convincing results based on hierarchical program representations. The overall system design and SFT+RL training pipeline are viewed as sound.

However, reviewers consistently raise concerns about the depth of evaluation and the strength of methodological justification. In particular, questions remain as to whether the contribution goes beyond integrating existing procedural and LLM-based components, and whether the experimental evidence sufficiently supports claims on scalability, efficiency, generalization, and the necessity of the multi-agent hierarchy. Additional concerns include limited scaling analysis, reliance on synthetic data, and partially justified design choices.

Although the rebuttal adds experiments, ablations, and clarifications that improve the paper, some reviewers remain unconvinced that the empirical evidence fully resolves these core concerns, leading to a borderline reject overall assessment.

**Reviewer Concerns:**

**Addressed Concerns**

- Lack of technical detail in procedural execution and asset handling (Raised by LEo5, xf2y):
Clarified program execution pipeline, asset sourcing (retrieval vs generation), and procedural assembly details.
- Missing efficiency and computational cost evaluation (Raised by LEo5, z19P, v4kQ):
Added inference time, memory usage, and multi-scale generation results.
- Absence of compliance and structural validity metrics for DSL outputs (Raised by v4kQ):
Introduced format accuracy, collision rate, PSA, and corresponding ablations.
- Insufficient ablation of RL vs SFT and reward design (Raised by v4kQ, xf2y):
Added PPO/DPO ablations and reward comparisons.
- Limited visual validation and lack of videos (Raised by z19P, xf2y):
Added demo videos and multi-step qualitative results.
- Clarification of training data construction and asset generation models (Raised by xf2y, z19P):
Detailed dataset generation process and committed to code release upon publication.

**Outstanding Concerns**

- Scalability and real-world feasibility for very large or complex cities (Raised by z19P):
While per-block scaling is shown, long-range scaling behavior and LLM cost implications remain only partially explored.
- Justification of the multi-agent hierarchical design (Raised by z19P):
Token-efficiency and performance gains are reported, but the necessity of each agent remains somewhat heuristic.
- Strength of novelty beyond integration of existing procedural and LLM components (Raised by LEo5)
Clarifications help, but some reviewers still view the contribution as incremental rather than fundamentally new.

**Reviewer Scores:**

- Reviewer LEo5 (4 → 6): This reviewer explicitly said “would not mind if accepted” and focused on missing details and experiments. Those were directly addressed.
- Reviewer v4kQ (6 → 6): Already positive but cautious. Rebuttal confirms rather than changes their view.
- Reviewer z19P (4 → 4): Concerns about scalability, LLM cost, and architectural necessity are only partially addressed. This reviewer is unlikely to move.
- Reviewer xf2y (4 → 4/6): Most concerns were clarity, transparency, and missing experiments/videos. These were comprehensively addressed.

---

### Decision · Program_Chairs · 2026-01-26

Reject